# YOLO-SAD: An Efficient SAR Aircraft Detection Network

Junyi Chen [1,2], Yanyun Shen [1,2] , Yinyu Liang [1,2], Zhipan Wang [1,2] and Qingling Zhang [1,2,*]

1 School of Aeronautics and Astronautics, Shenzhen Campus of Sun Yat-Sen University, No. 66, Gongchang Road, Guangming District, Shenzhen 518107, China; chenjy689@mail2.sysu.edu.cn (J.C.); shenyy28@mail2.sysu.edu.cn (Y.S.); liangyy255@mail2.sysu.edu.cn (Y.L.); wangzhp25@mail2.sysu.edu.cn (Z.W.)
2 Shenzhen Key Laboratory of Intelligent Microsatellite Constellation, Shenzhen Campus of Sun Yat-Sen University, No. 66, Gongchang Road, Guangming District, Shenzhen 518107, China
* Correspondence: zhangqling@mail.sysu.edu.cn

**Abstract:** Aircraft detection in SAR images of airports remains crucial for continuous ground observation and aviation transportation scheduling in all weather conditions, but low resolution and complex scenes pose unique challenges. Existing methods struggle with accuracy, overlapping detections, and missed targets. We propose You Only Look Once-SAR Aircraft Detector (YOLO-SAD), a novel detector that tackles these issues. YOLO-SAD leverages the Attention-Efficient Layer Aggregation Network-Head (A-ELAN-H) module to prioritize essential features for improved accuracy. Additionally, the SAR Aircraft Detection-Feature Pyramid Network (SAD-FPN) optimizes multi-scale feature fusion, boosting detection speed. Finally, Enhanced Non-Maximum Suppression (EH-NMS) eliminates overlapping detections. On the SAR Aircraft Detection Dataset (SADD), YOLO-SAD achieved 91.9% AP(0.5) and 57.1% AP(0.5:0.95), surpassing the baseline by 2.1% and 1.9%, respectively. Extensive comparisons on SADD further demonstrate YOLO-SAD's superiority over five state-of-the-art methods in both AP(0.5) and AP(0.5:0.95). The outcomes of further comparative experiments on the SAR-AIRcraft-1.0 dataset confirm the robust generalization capability of YOLO-SAD, demonstrating its potential use in aircraft detection with SAR.

**Keywords:** synthetic aperture radar; space borne remote sensing; aircraft detection; convolutional neural network; non-maximum suppression

## 1. Introduction

Aircraft are pivotal geographic targets, essential for civilian transport and disaster response. SAR aircraft detection stands out due to its ability to operate effectively through clouds and varying lighting, a significant advantage over other detection methods. This capability is particularly important for covert detection, tracking, and counter-terrorism, even in harsh weather [1]. However, SAR imagery often contains noise and interference, and the complex scenes at airports present considerable challenges for accurate target detection, even with advanced deep learning models. A simple strategy to improve accuracy is to increase network parameters, but this could slow down model detection speed. Therefore, achieving a balance between precision and speed is key for effective and reliable aircraft detection in SAR remote sensing scenarios.

Researchers have developed diverse techniques to address the challenges of SAR aircraft detection. Notably, anchor-based one-stage object detection methods have achieved a balance between detection precision and real-time performance. For example, the Salient Fusion Module (SFM) integrates high-resolution details with semantic features to enhance feature discrimination in complex scenes [2]. Zhang et al. introduced the SAR Aircraft Detection Dataset (SADD) and the SEFEPNet, a deep learning model leveraging domain adaptive transfer learning for better detection [3]. Additionally, Han et al.'s Low-Level Semantic Enhancement Module (LSEM) focuses on enhancing scattered features in SAR imagery [4]. Zhang et al.'s Fusion Local and Contextual Attention Pyramid (FLCAPN) further

advances detection by merging local and contextual features. Each of these approaches contributes to the ongoing improvement of SAR aircraft detection capabilities [5].

Despite advancements, current SAR aircraft detection methods face challenges in achieving high accuracy, resolving overlapping detections, minimizing missed targets, and reducing false positives. Firstly, one-stage anchor-based detection methods, originally designed for natural images, may not align well with the unique characteristics of SAR imagery, leading to suboptimal accuracy. Secondly, these methods produce numerous candidate bounding boxes at different scales, which can result in unresolved overlaps during post-processing, degrading the clarity and precision of the detection results.

To overcome the challenges associated with SAR imagery, we introduce a tailored feature pyramid structure that de-emphasizes high-level features during fusion, aligning with the low-resolution nature of SAR images. This design not only simplifies the model but also lessens the likelihood of overfitting. Furthermore, to address the issue of overlapping detections, we have developed an Enhanced Non-Maximum Suppression (EH-NMS) technique for post-processing. EH-NMS calculates the union-to-max area ratio for candidate bounding boxes, enabling precise elimination of fully overlapping small boxes, thereby refining the detection output for SAR aircraft detection.

This paper introduces several key contributions:

1. We present YOLO-SAD, a cutting-edge network for SAR aircraft detection, which includes the SAR Aircraft Detection-Feature Pyramid Network (SAD-FPN). This feature fusion network is customized for SAR imagery, adept at harnessing the rich texture information while filtering out less valuable data. Additionally, the Attention-Efficient Layer Aggregation Network-Head (A-ELAN-H) module integrates channel and spatial attention mechanisms to highlight critical information from SAR image feature maps.
2. To tackle the issue of overlapping detection boxes in SAR aircraft detection, we introduce the Enhanced Non-Maximum Suppression (EH-NMS) post-processing technique. This method significantly refines the detection accuracy and enhances the clarity of the output visualization.

## 2. Related Work

### 2.1. Object Detection Networks

Deep learning-based object detection networks generally fall into two categories: one-stage and two-stage detectors [6]. Pioneered by R-CNN in 2014, two-stage detectors mimic human object recognition by splitting the process into distinct stages. Initially, regions of interest (ROIs) are identified, followed by classification and bounding box refinement [7]. Fast R-CNN improved upon this by integrating the Region Proposal Network (RPN) and R-CNN, enhancing detection speed [8]. Further advancements led to the development of Faster R-CNN in 2015, which introduced the ROI pooling layer for faster mapping of candidate regions to feature maps [9]. Mask R-CNN, introduced in 2017, expanded on Faster R-CNN by adding pixel-level segmentation, providing more detailed object information. Despite their complexity, two-stage detectors are favored for their high accuracy and robustness in object detection tasks [10].

One-stage object detection networks streamline the process by directly predicting bounding boxes and class probabilities in a single operation, eliminating the need for a separate object localization step. The You Only Look Once (YOLO) network, introduced by Joseph Redmon and Ali Farhadi in 2015, was a trailblazer in this approach. YOLO operates by dividing the image into grids and predicting classes and bounding boxes for each grid cell, which simplifies detection and accelerates speed [11].

The Single Shot Multi-Box Detector (SSD), also proposed in 2015 by Wei Liu et al., enhances one-stage detection by utilizing multi-scale convolutional features to detect objects of varying sizes, boosting accuracy [12]. RetinaNet, introduced by Tsung-Yi Lin et al., addresses class imbalance with the Focal Loss function, which assigns greater weight to difficult cases, leading to improved precision [13]. One-stage detectors are valued for

their speed and simplicity, offering real-time detection capabilities that are particularly advantageous in fast-paced environments [6].

### 2.2. SAR Remote Sensing Object Detection Methods

SAR remote sensing object detection techniques generally divide into manual feature-based and deep learning-based strategies.

Manual feature-based methods for SAR object detection harness a variety of SAR data characteristics, including polarization, statistical, texture, shape, and size features, to identify objects. For instance, CFAR-based methods use SAR backscatter to separate targets from clutter [14–16], while polarization-based methods analyze polarimetric SAR data for target signatures [17–20]. Geometric feature-based methods rely on dimensions and proportions [21–25], and HOG-based methods focus on local gradients for geometric invariance in detection [26–28]. Haar-feature-based methods identify targets through grayscale patterns using templates [29–31], and SIFT operator-based methods detect objects by matching invariant feature points across noise and transformations [32,33].

Despite their straightforward implementation, these manual feature-based methods face challenges due to the diversity and complexity of SAR image targets, which can lead to reduced accuracy and robustness.

Since 2010, deep learning has revolutionized object detection, with significant advancements in SAR remote sensing. These methods automate feature learning, greatly improving the accuracy and efficiency of SAR target detection. Notable developments include:

- Guo et al.'s Scattering Information Enhancement (SIE) method, which uses the Harris–Laplace detector and DBSCAN for preprocessing to enhance aircraft scattering and employs GMM for modeling [34].
- Yang et al.'s R-RetinaNet, introducing Rotatable Bounding Boxes (RBoxes) to address feature scale mismatch in SAR detection tasks [35].
- Xu et al.'s strategy for on-board SAR ship detection, combining CFAR for initial detection with YOLOv4 [36] for refined results and faster detection speeds [37].
- Yu et al.'s modification of the YOLOX-s structure, eliminating the computationally heavy pyramidal framework for quicker detection [38].
- Wang et al.'s Semantic Condition Constraint Guided Feature Aware Network (SCFNet), which integrates semantic condition constraints to boost the network's discriminative capabilities for multi-label aircraft detection in SAR images [39].

These contributions showcase the potential of deep learning in enhancing SAR object detection across various aspects.

### 2.3. Post-Processing Methods for Object Detection

Post-processing in object detection involves refining algorithm outputs to boost accuracy and reliability. Common techniques include Non-Maximum Suppression (NMS) [40], thresholding, bounding box regression, non-linear transformations, and result fusion.

The threshold method classifies detection outputs as positive or negative using confidence scores or Intersection over Union (IoU) as criteria, filtering out low-confidence results while occasionally leading to missed detections. Bounding box regression fine-tunes box coordinates for greater precision, often using regression models to predict coordinate offsets. Non-linear transformations, such as scaling and shape adjustments, are applied to bolster detection robustness, typically implemented via Convolutional Neural Networks (CNNs). Fusion techniques amalgamate various detection outputs, often using weighted averaging or voting, to enhance overall accuracy.

Non-Maximum Suppression (NMS) is a standard post-processing technique in anchor-based object detection that filters overlapping candidate regions to retain the most probable object regions. Despite its effectiveness, NMS can result in missed detections [41]. To overcome this, researchers have introduced several refined NMS variants:

-  Soft-NMS, introduced by Bodla et al., mitigates the issue by reducing confidence scores of overlapping boxes rather than eliminating them, thus improving overall detection accuracy [42].
-  Adaptive NMS, proposed by Liu et al., is an iterative algorithm that dynamically adjusts the suppression threshold based on the dimensions and aspect ratios of candidate regions, enhancing detection efficiency [43].
-  DIoU-NMS, developed by Zheng et al., employs the Distance-IoU (DIoU) metric in place of IoU, refining the suppression process and further improving accuracy [44].
-  Truncated NMS, presented by Shen et al., addresses the challenge of overlapping regions during result merging in remote sensing image detection [45].

These methods represent targeted efforts to refine NMS, particularly in addressing the persistent issue of overlapping detections.

The methods mentioned earlier are predominantly designed for natural images and do not account for the distinct geometric and positional traits of remote sensing targets. This limitation can lead to the problem of overlapping detection boxes.

## 3. Materials and Methods

This study introduces a refined approach for boosting the precision and clarity of SAR aircraft detection. It integrates the YOLO-SAD network, an efficient one-stage anchor-based detection system, with the Enhanced Non-Maximum Suppression (EH-NMS) post-processing technique. YOLO-SAD, detailed in Figure 1, consists of a backbone for feature extraction, a feature fusion pyramid network for multi-scale feature integration, and a detection head for inference and result generation. EH-NMS is applied to polish YOLO-SAD's outputs, enhancing the visual quality of detected aircraft.

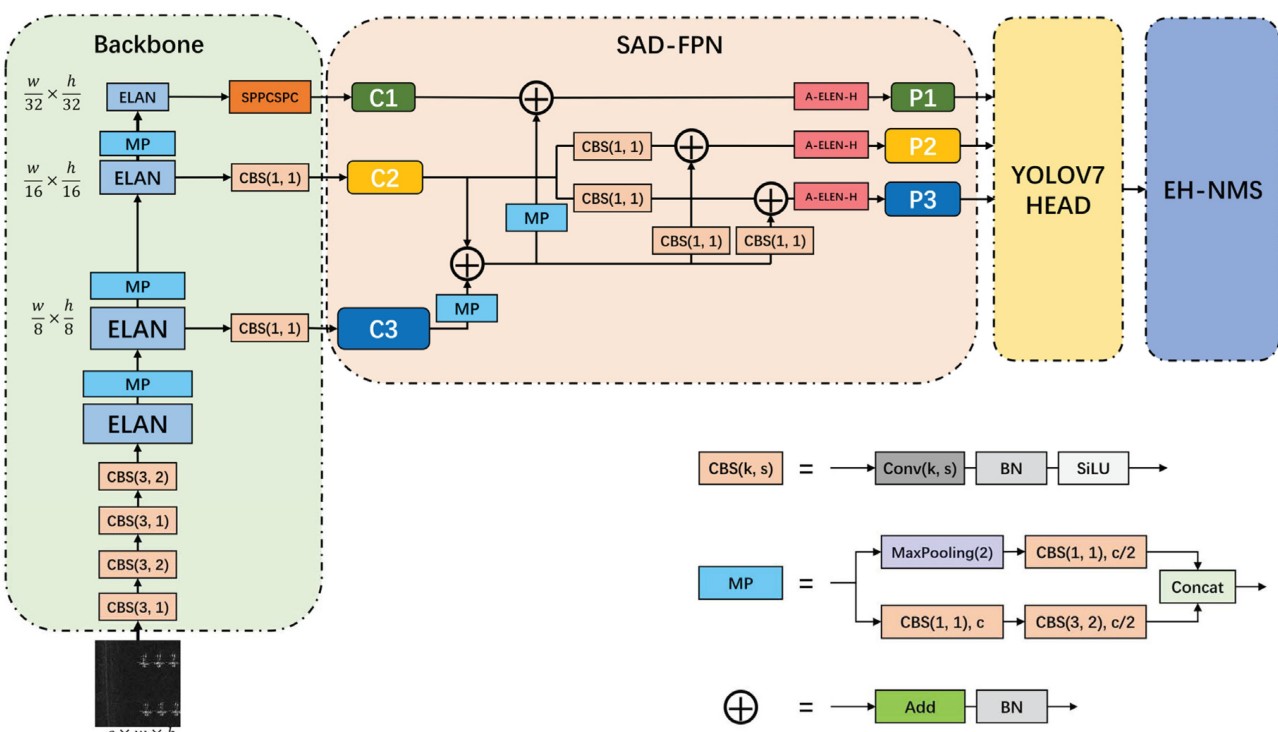

**Figure 1.** Architecture of YOLO-SAD network. Conv(k, s) means convolution with k × k kernel size and stride is s; BN represents batch normalization; SiLU represents SiLU activate function; MP means max-pooling module; MaxPooling(k) represent Max pooling with a kernel size of k; + represents element-wise addition followed by batch normalization; and Add represents element-wise addition. C1, C2 and C3 represent the input feature maps of SAD-FPN. Similarly, P1, P2 and P3 represent the output feature maps of SAD-FPN.

### 3.1. The Network Structure of YOLO-SAD

#### 3.1.1. Backbone Network

The backbone network in YOLO-SAD extracts multi-scale features from inputs for object detection, prioritizing accuracy while reducing computational and memory demands. It comprises Convolution Batch-Normal SiLu (CBS) modules for feature extraction, Max pooling (MP) modules for down-sampling, and Efficient Layer Aggregation Network (ELAN) [46] modules. The CBS(k, s) module includes a convolutional layer with a kernel size of n and stride m, followed by batch normalization and SiLU [47] activation for non-linear transformations. The MP module merges CBS(3, 2) and pooling layers to achieve 8×, 16×, and 32× down-sampling, generating feature maps of the 1/8, 1/16 and 1/32 input image's size, respectively. This process enlarges the receptive field and decreases feature map dimensions.

As depicted in Figure 2, the ELAN module blends concepts from VoVNet [48] and CSPNet [49]. It splits base layer feature maps into long and short paths, with the long paths further processed by CBS modules before merging back through cross-stage connections. The ELAN module's long paths integrate VoVNet's connection pattern and employ One-Shot Aggregation (OSA), which consolidates outputs from various CBS modules, maintaining intermediate feature information. This design enables the ELAN module to capture long-range feature dependencies efficiently.

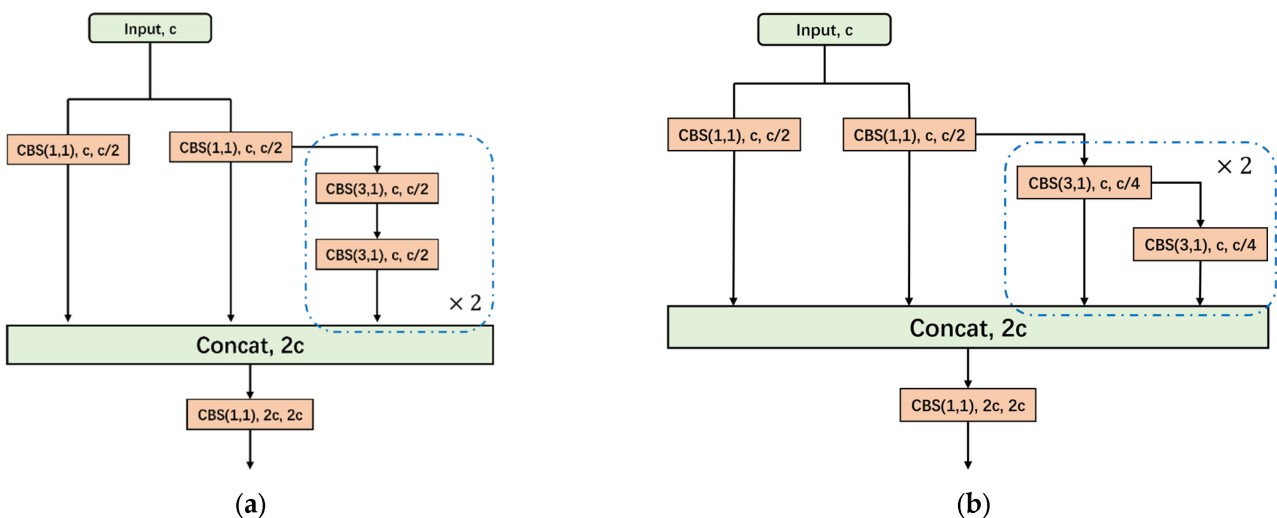

**Figure 2.** (**a**) ELAN module. (**b**) ELAN-H module. CBS, c, c/2 means the input channel number is c and the output channel number is c/2.

#### 3.1.2. SAD-FPN for SAR Image Feature

The feature fusion network in SAD-FPN merges feature maps from multiple layers using up-sampling and down-sampling to enhance feature representation, which is crucial for effective object detection. Drawing on the Path Aggregation Network (PAN) [50] and considering the peculiarities of SAR imagery, we have developed the SAD-FPN to refine feature fusion for SAR images.

The conventional PAN, as shown in Figure 3, features a top-down and bottom-up Feature Pyramid Network (FPN). The top-down pathway up-samples to infuse lower-level features with semantic cues, while the bottom-up pathway down-samples to add spatial details to higher-level features. However, SAR images' low resolution and pervasive coherent noise can compromise higher-level features, making heavy reliance on them during fusion detrimental to detection performance. SAD-FPN addresses these challenges, tailoring the fusion process for the unique demands of SAR remote sensing.

We evaluated the influence of multi-scale features on detection precision by visualizing Grad-CAM masks of the traditional PAN's input (C3, C2, C1) and output (O3, O2, O1)

feature maps, as shown in Figure 4. Detection accuracy is directly related to the quality of information in C1, C2, and C3. Our Grad-CAM analysis indicates that C1 holds more noise and inaccuracies than C2 and C3, and up-sampling C1 could transfer these defects to higher levels, negatively affecting the fusion quality.

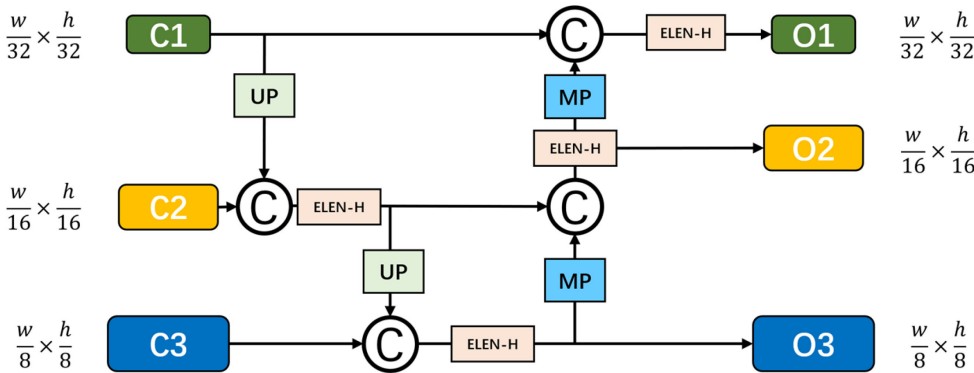

**Figure 3.** Architecture of PAN utilized in the baseline. C1, C2, and C3 represent the inputs of PAN; O1, O2, and O3 represent the outputs of PAN; UP represents nearest up-sampling with a factor of 2; and C represents feature concatenation.

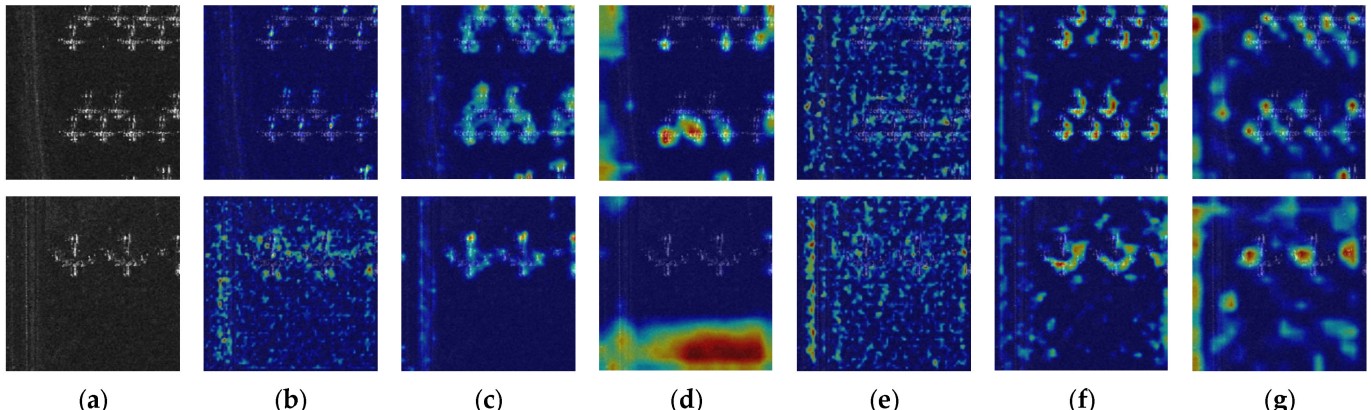

**Figure 4.** Grad-CAM masks of different scale feature maps of the baseline: (**a**) input image; (**b**) C3; (**c**) C2.; (**d**) C1; (**e**) O3; (**f**) O2; and (**g**) O1. The brighter the color, the higher the importance of that location for the aircraft.

To address the propagation of coherent noise, we have developed SAD-FPN, an adaptation of the standard PAN that minimizes reliance on high-level features during fusion. This innovation significantly improves network accuracy, as evidenced by the ablation study in Section 4.2.4.

Furthermore, Figure 4 displays Grad-CAM masks for O1, O2, and O3, which serve as both fusion network outputs and inputs to the detection head. O2's masks (Column (f)) concentrate on aircraft regions, bolstering instance detection, whereas O3's masks (Column (e)) distribute attention broadly, offering less detection benefit. Reassigning O2 as the input to the head network not only enhances its impact but also streamlines the network by reducing the feature map size, optimizing it for efficient aircraft detection in SAR imagery. The success of this approach is supported by the ablation study in Section 4.2.4.

The SAD-FPN architecture, as illustrated in Figure 5, starts with C1, C2, and C3, which are down-sampled feature maps from the backbone network by factors of 32×, 16×, and 8×, respectively. These maps are independently processed by CBS modules and the SPPCSPC module (Figure 6) before being input into SAD-FPN.

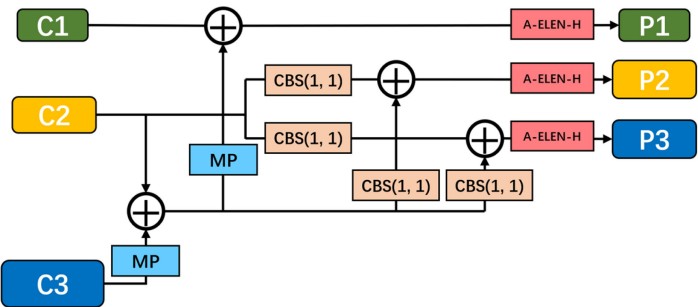

**Figure 5.** Architecture of specified SAD-FPN designed for SAR image features.

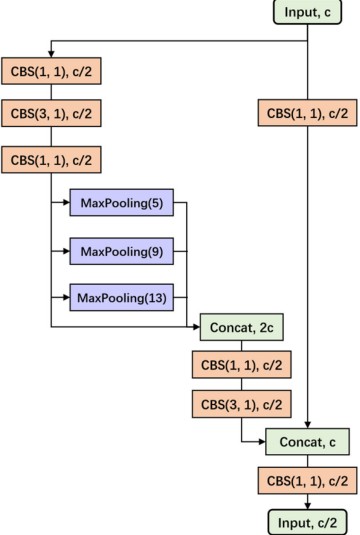

**Figure 6.** Architecture of SPPCSPC block. Unless otherwise specified, the output channel number of CBS block is c. MaxPooling(k) represents Max pooling with a kernel size of k.

To ensure feature compatibility, C3's output from the MP module and C2 are fused using feature addition, aligning the intermediate feature C2 with the low-level feature C3. The combined features then pass through MP and CBS modules and are fused with C1 and C2 through additional feature addition, enabling multi-scale feature integration.

Finally, the A-ELAN-H module, detailed in Section 3.1.3, encodes the fused features to produce SAD-FPN's final outputs, P1, P2, and P3.

### 3.1.3. Attention Feature Encoding Module A-ELAN-H

Our proposed A-ELAN-H module integrates attention mechanisms into neural networks, enabling selective focus on essential input information, which enhances performance and generalization without a substantial increase in parameters or complexity. We posit that attention mechanisms, when incorporated into the YOLO network, can elevate the efficacy of object detection in SAR images.

While self-attention and multi-head attention [51] are prevalent in sequence detection, they often involve a substantial number of model parameters. In contrast, channel and spatial attention mechanisms [52,53] are more appropriate for SAR aircraft detection, offering the benefits of attention without a significant rise in model complexity or detection costs.

Object detection aims to identify targets and their locations, but not all channels and features are equally important for this task. To focus on the most relevant information, we have implemented channel attention to highlight crucial channels for recognition, leveraging inter-channel relationships. This integration with the ELAN-H module strengthens the model's ability to identify targets by concentrating on significant channels. Likewise, spatial attention is used to pinpoint areas where instances are probable to occur, utilizing

inter-spatial relationships to refine localization. The value of these approaches is validated by the ablation study presented in Section 4.2.2.

The A-ELAN-H module, as detailed in Figure 7, integrates the Convolutional Block Attention Module (CBAM) [53] with the ELAN module to refine feature representation. The process begins with the channel attention module, which identifies and emphasizes significant features. It uses global max and average pooling to generate channel descriptor vectors, which are then down-sampled and up-sampled back to the original channel dimension using fully connected layers. Sigmoid activation function is applied to these vectors to generate channel weights, which are multiplied element-wise with the input feature map.

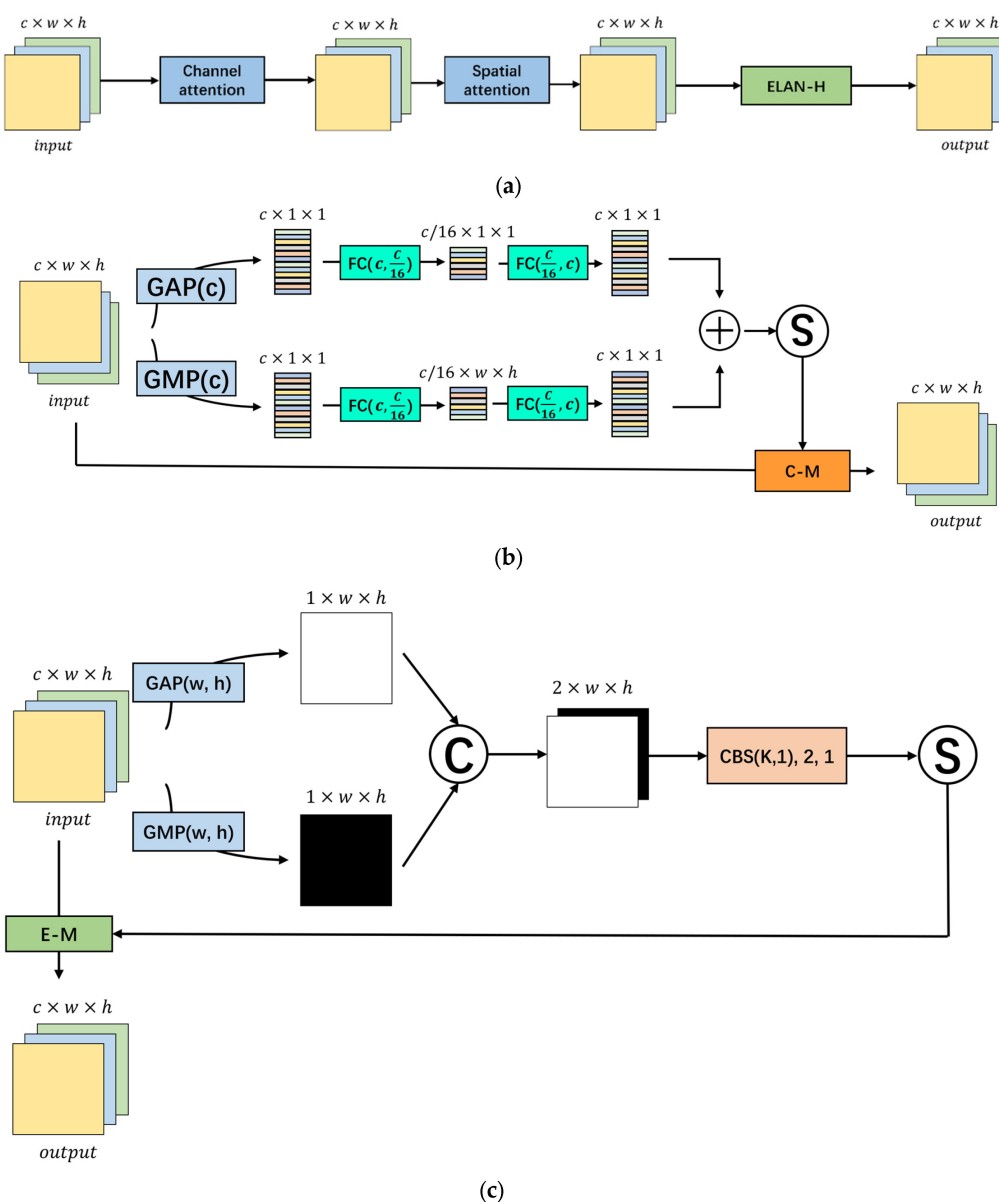

**Figure 7.** (**a**) A-ELAN-H module. (**b**) Channel attention module. (**c**) Spatial attention module. GAP(c/(w, h)) means global average pooling along with the direction of channel or (width, high). GMP(c/(w, h)) means global maximum pooling along with the direction of channel or (width, high). S represents Sigmoid activate function. + represents element-wise addition. FC(a, b) represents full connected layer with input size of a and output size of b. C represent concatenation. C-M represents channel-wise multiplication. E-M represents element-wise multiplication. The colors of the vectors in (**b**) correspond to different elements within these vectors.

Following the channel attention module, the feature map enters the spatial attention module, which highlights relevant spatial locations. This module applies global average and max pooling operations across channels, resulting in a reduced-channel feature map. After passing through the CBS module and a sigmoid activation, it produces spatial weights that are multiplied with the input feature map to emphasize key spatial areas.

The feature map, now enhanced with both channel and spatial attention, is then passed to the ELAN-H module. The ELAN-H module builds upon the ELAN by effectively utilizing intermediate feature layer information for feature encoding, thereby improving the overall feature representation for object detection tasks.

### 3.1.4. Detection Head

The YOLO-SAD detection head performs object detection using the feature fusion network's output [54]. It produces three candidate bounding boxes with varying aspect ratios at each position on the feature maps, which are down-sampled to 1/16 and 1/32 of the input image's size. For each box, the head determines the object's detection results, providing the object's center coordinates (x, y), height (h), and width (w), as well as confidence scores for the object's presence and predicted class.

### 3.2. Loss Function

The method uses the same three-component loss function as the baseline for network training, consisting of coordinate loss, object confidence loss, and classification loss [54]:

$$L = \lambda_1 L(IOU) + \lambda_2 L(OBJ) + \lambda_3 L(CLS) \tag{1}$$

The loss function comprises three components: coordinate loss, object confidence loss, and classification loss, with respective weights of 0.05, 0.7, and 0.3 [54]. The coordinate loss assesses the detection box's positional accuracy relative to the target box, calculated as the mean error between the IoU of all positive predicted boxes with the true boxes and 1. The object confidence loss evaluates the detection box's likelihood of containing an object, using binary cross-entropy. Lastly, the classification loss measures the difference between the predicted class confidence and the actual label, also applying binary cross-entropy.

### 3.3. Area-Based EH-NMS Post-Processing Method

Non-Maximum Suppression (NMS) is a standard post-processing technique in anchor-based object detection that refines the selection of the most accurate bounding box from a set of candidates. During detection, multiple candidate boxes with various aspect ratios are generated, and each is assigned a confidence score. NMS addresses the issue of multiple boxes for a single object by initially selecting the box with the highest score. It then calculates the Intersection over Union (IoU) between this box and the others, discarding those with an IoU above a set threshold. This cycle continues, iteratively removing lower-ranked boxes with overlapping predictions, until only the most confident, non-overlapping box remains for each object.

Figure 8 highlights a common issue with anchor-based one-stage object detection networks when applied to the SADD dataset: the mistaken exclusion of overlapping detection boxes, which significantly lowers the precision metric during evaluation.

The persistence of overlapping detection boxes after NMS in SAR imagery is due to the sparse target features that can mislead the object detection network. The network may assign high confidence to partial aircraft detections, treating them as complete objects. This leads to a size discrepancy between the smaller detected parts (m1) and the actual larger aircraft (m2). When calculating IoU, m1 defines the intersection area and m2 the union area, resulting in a low IoU that may be below the threshold, failing to eliminate the overlapping boxes. Simply adjusting the IoU threshold is not a viable solution, as it can lead to the removal of correctly detected adjacent aircraft, thus lowering the recall rate. This issue is particularly prevalent due to the typical parking patterns of aircraft on the ground.

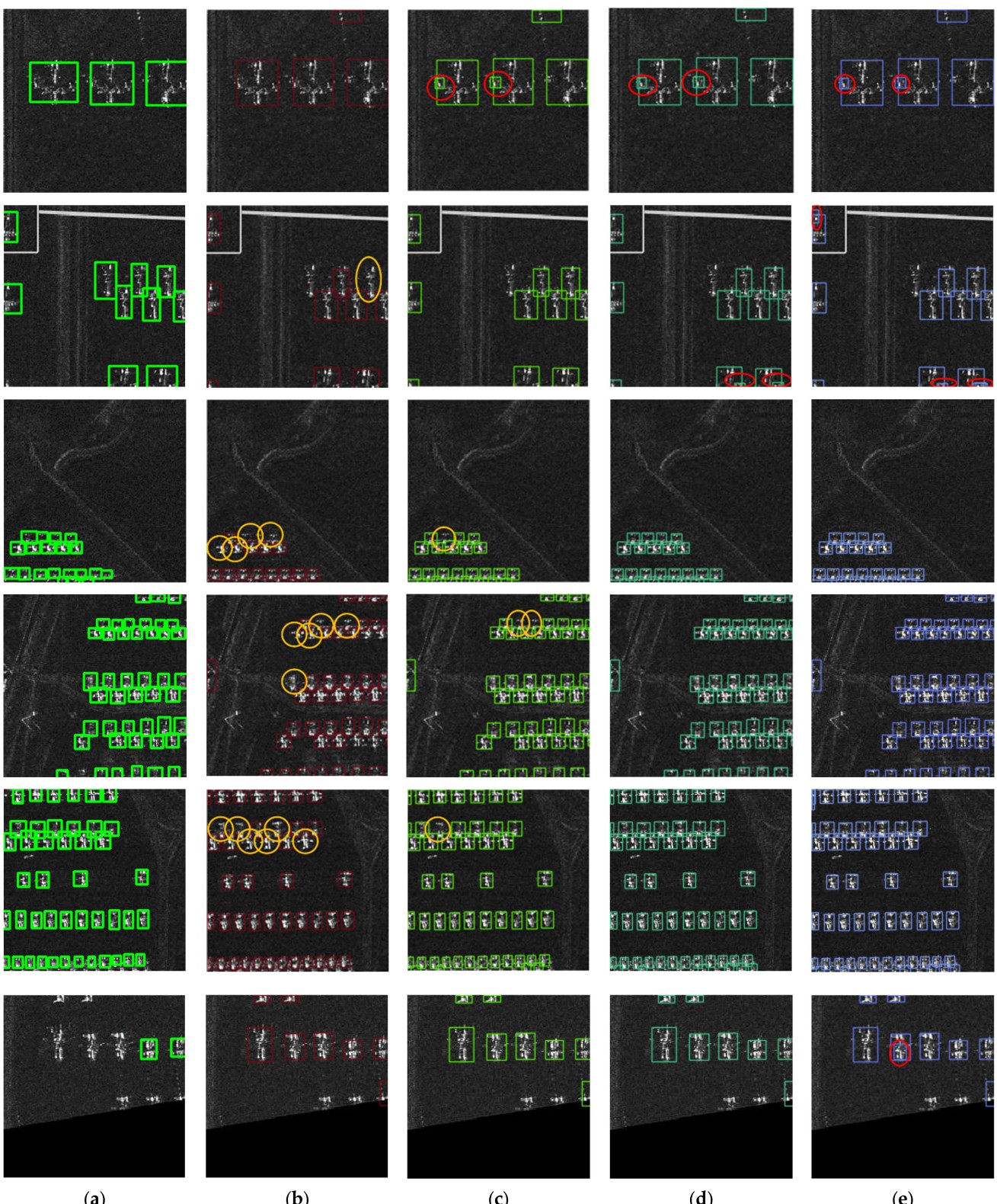

**Figure 8.** Typical detection box errors coursed by NMS method under different IOU thresholds. The red lines indicate the erroneous overlapping detection boxes. The yellow lines indicate erroneous missing detect boxes. (**a**) Ground truth; (**b**) 0.025; (**c**) 0.05; (**d**) 0.1; and (**e**) 0.4. The green rectangles represent the ground true labels, while the rectangles of other colors correspond to the predicted bounding boxes from different IOU threshold; The yellow circles indicate mistakes eliminated, while the red circles denote overlapped bounding boxes.

We introduce an Enhanced Non-Maximum Suppression (EH-NMS) algorithm customized for SAR aircraft detection, which operates in three steps. First, a threshold $\theta$ for the area ratio is established, and the process begins for each image. The box with the highest score is chosen as the master (M), and the Intersection over Union (IoU) is computed for M against the rest of the candidates (B). As with traditional NMS, boxes in B with IoU above the threshold are removed.

In the third step, EH-NMS calculates a ratio $\mu$ for each box in B, comparing the union area with M's area. A $\mu$ below $\theta$ indicates a significant overlap with M, prompting exclusion, while a $\mu$ above $\theta$ suggests minimal overlap, allowing retention. This evaluation repeats until all overlaps are addressed. EH-NMS effectively reduces overlapping detections. Algorithm 1 provide pseudocode detailing its implementation.

---

**Algorithm 1** EH-NMS

---

        **Input: B** = [b1,..., bn], **S** = [s1,..., sn], **IoUt**, $\theta$
        **B** is the list of all candidate detection boxes
        **S** is the list of scores corresponding to all candidate boxes
        **IoUt** is the IoU threshold
        $\theta$ is the threshold of the ratio of the parallel area to the maximum
         possible detection frame area
        **Output: K**
        **K** indicates the detection boxes retained after the EH-NMS

| | | |
|---|---|---|
| 1 | : | **for** image **in** images |
| 2 | : | K $\leftarrow$ [ ] |
| 3 | : | **while** B $\neq \varnothing$ **do**: |
| 4 | : | m $\leftarrow$ argmax(S) |
| 5 | : | M $\leftarrow$ bm |
| 6 | : | K $\leftarrow$ K $\cup$ M |
| 7 | : | B $\leftarrow$ B—bm; S $\leftarrow$ S—sm |
| 8 | : | **for** bi in B **do**: |
| 9 | : | **if** IoU(M, bi) < IoUt: |
| 10 | : | B $\leftarrow$ B—bi; S $\leftarrow$ S—si |
| 11 | : | **else**: |
| 12 | : | B $\leftarrow$ B; S $\leftarrow$ S |
| 13 | : | **if** ratio(union(M, bi), area(M)) > $\theta$: |
| 14 | : | B $\leftarrow$ B; S $\leftarrow$ S |
| 15 | : | **else**: |
| 16 | : | B $\leftarrow$ B—bi; S $\leftarrow$ S—si |
| 17 | : | **return** K |
| 18 | : | **end** |

---

## 4. Experiment and Results

### 4.1. Dataset and Evaluation Metrics Introduction

#### 4.1.1. Dataset Introduction

The SADD is a publicly available dataset for SAR aircraft detection, comprising 2966 224 × 224 pixel images [3]. Noting that the dataset had been augmented for diversity, we conducted a fresh split to prevent duplication between training and validation sets. We meticulously selected 701 unique images, forming a 60:40 training-to-validation ratio. The training set received further augmentation through techniques like rotation and cropping.

To validate our method's robustness, we also conducted experiments on the SAR-AIRcraft-1.0 dataset, which includes 4368 images from the Gaofen-3 satellite with 1 m × 1 m resolution and 16,463 annotated instances in four sizes [55]. This dataset was divided into an 80:20 training-to-validation ratio, with 3499 images and 13,812 instances for training, and 869 images with 2651 instances for validation.

### 4.1.2. Experimental Setup

All network training for this study was performed on an Nvidia RTX 3090 GPU using PyTorch 1.7.1 with a batch size of 20. Performance evaluations were carried out on an Nvidia Tesla V100-PCIE-32GB GPU with PyTorch 1.13.1 and a batch size of 32. Given the SADD's high resolution and the network's parameter efficiency, we trained for 300 epochs to ensure a complete learning cycle. For the SAR-AIRcraft-1.0 dataset, which has a similar resolution but double the scale of SADD, we trained for 400 epochs to prevent overfitting. As SAR and natural images differ significantly, we avoided using pretrained models for both datasets.

We trained the network using the Stochastic Gradient Descent (SGD) [56] optimizer with an initial learning rate of 0.01, reduced cyclically by a factor of 0.1, a momentum of 0.937, and a weight decay of 0.0005. A 3-epoch warm-up phase was implemented, starting with a momentum of 0.8 and a bias learning rate of 0.1. Detailed hyperparameters are presented in Table 1.

**Table 1.** Hyperparameters setting for training.

| Hyperparameters | Value |
| --- | --- |
| Optimizer | SGD |
| Initial learning rate | 0.01 |
| Cyclic learning rate decay | 0.1 |
| Momentum | 0.937 |
| Weight decay | 0.0005 |
| Batch size | 20 |
| Epochs | 300 |

### 4.1.3. Evaluation Metrics

This study employs standard deep learning evaluation metrics: accuracy, recall, and precision. To clarify these metrics, we define key terms: a False Negative (FN) is when the actual label is positive, but the model predicts negative; a False Positive (FP) is when the actual label is negative, but the model predicts positive; and a True Positive (TP) is when both the actual and predicted labels are positive.

Precision measures the model's accuracy in identifying positive instances among all predicted positives, calculated as the ratio of TP to the sum of TP and FP.

$$\text{precision} = \frac{\text{TP}}{\text{TP} + \text{FP}} \tag{2}$$

Recall measures the model's effectiveness in identifying all actual positive samples, reflecting its target recognition capability. Mathematically, recall is computed as the ratio of True Positives (TP) to the sum of True Positives and False Negatives (FN):

$$\text{recall} = \frac{\text{TP}}{\text{TP} + \text{FN}} \tag{3}$$

For a thorough assessment of a detection algorithm's ability to handle negative samples and retrieve targets, robust metrics are crucial. Average Precision (AP) is the area under the precision–recall curve, with precision on the *y*-axis and recall on the *x*-axis. A greater area under this curve signifies higher detection accuracy. AP(0.5) is the AP at an IoU threshold of 0.5, while AP(0.5:0.95) is the mean of AP values across IoU thresholds from 0.5 to 0.95:

$$\text{AP} = \int_0^1 \text{p(r)d}_\text{r} \tag{4}$$

$$\text{AP}(0.5 : 0.95) = \frac{\sum_{\text{i}=0.5}^{0.95} \text{AP(i)}}{10} \tag{5}$$

Frames Per Second (FPS) is a standard measure of a network's speed and efficiency, indicating how many images the network can process in a second. A higher FPS signifies a faster object detection network with a greater frame rate.

$$\text{FPS} = \frac{\text{N}}{\text{t}} \quad (6)$$

### 4.2. Ablation Experiment

4.2.1. Ablation Experiment on Feature Fusion Methods

Our SAD-FPN architecture for SAR image features involves merging features across different scales, which can be achieved through direct or indirect fusion techniques. Direct fusion methods commonly used in deep learning include addition, multiplication, and concatenation of features. The YOLOR study suggests that concatenation and addition generally yield better results than multiplication, though concatenation can increase the number of weight parameters [57].

Indirect fusion methods, introduced in YOLOv4 [36], involve combining feature concatenation with $1 \times 1$ convolutions instead of relying solely on direct fusion. This strategy allows the network to learn effective feature fusion autonomously. We assessed the performance of various direct and indirect fusion methods in our feature fusion module through ablation experiments, with the results summarized in Table 2.

**Table 2.** Ablation study on different feature fusion methods.

| Method | AP(0.5) | AP(0.5:0.95) | Precision | Recall | Params (M) | Flops (G) | FPS |
|---|---|---|---|---|---|---|---|
| Concat + Conv | 89.6% | 55.7% | 88.9% | 82.7% | 71.5 | 98.5 | 42.9 |
| Concat | 89.9% | 54.7% | 95.2% | 80.1% | 71.7 | 98.9 | 42.2 |
| Add | 91.5% | 56.0% | 91.5% | 87.4% | 69.9 | 97.2 | 43.7 |
| Add + SiLU | 90.5% | 55.1% | 90.2% | 84.2% | 69.9 | 97.2 | 43.7 |
| Add + BN | 92.0% | 56.3% | 94.2% | 83.3 | 69.9 | 97.2 | 43.7 |
| Add + BN + SiLU | 89.3% | 55.3% | 95.6% | 80.1% | 69.9 | 97.2 | 43.7 |

Analysis of Table 2 leads to the following conclusions:

1. Feature concatenation compared to addition results in lower FPS by 0.8, a 1.3 GB increase in model parameters, and a 0.6 MB increase in model size, due to doubling the channel number.
2. Indirect fusion, already slower than concatenation, is further hindered by additional convolutional layers, leading to a 0.7 FPS decrease and a 0.4 GB parameter increase. Despite a 0.3% improvement in AP(0.5) and a 6.3% rise in precision, there is a 1% drop in AP(0.5:0.95) and a 2.6% decrease in recall.
3. Feature addition-based fusion outperforms both indirect fusion and concatenation in accuracy and speed. While SiLU activation and batch normalization do not add complexity, SiLU negatively affects the addition method, lowering AP(0.5) by 1% and AP(0.5:0.95) by 0.9%. However, batch normalization improves these metrics by 0.5% and 0.3%, reaching 92.0% and 56.3%, respectively.
4. Combining SiLU and batch normalization in feature addition reduces primary accuracy metrics compared to using batch normalization alone, with a 2.7% and 1% decrease in AP(0.5) and AP(0.5:0.95).

In summary, feature addition with batch normalization is the most effective fusion method in terms of accuracy and speed, as evidenced by the highest AP(0.5) and AP(0.5:0.95) values. Therefore, we select this combination as the optimal feature fusion strategy for our network.

4.2.2. Ablation Experiment on the A-ELAN-H Module

The backbone network's extracted features are crucial for further processing. To prevent the random initialization of attention mechanism weights from interfering with

the backbone's weights and degrading prediction accuracy, this study applies the attention feature encoding module only to the network's neck. Given the need for rapid detection, self-attention and multi-head attention mechanisms are excluded. Ablation experiments were conducted to evaluate the effects of different attention mechanisms in the encoding module and kernel size variations in the spatial attention mechanism, with results presented in Table 3.

**Table 3.** Ablation study on the A-ELAN-H module. $\sqrt{}$ indicates that this attention mechanism is adopted; - indicates that this attention mechanism is excluded.

| Channel Attention | Space Attention | AP(0.5) | AP(0.5:0.95) | Precision | Recall | Params (M) | Flops (G) | FPS |
|---|---|---|---|---|---|---|---|---|
| - | - | 89.8% | 55.2% | 92.6% | 82.7% | 74.8 | 105.1 | 40.3 |
| - | (5, 5, 5, 5) | 91.1% | 54.6% | 90.2% | 82.0% | 71.3 | 105.1 | 42.9 |
| $\sqrt{}$ | - | 91.4% | 56.2% | 91.9% | 83.4% | 71.7 | 105.5 | 40.9 |
| $\sqrt{}$ | (5, 7, 5, 3) | 92.2% | 55.8% | 88.1% | 89.7% | 71.7 | 105.5 | 40.3 |
| $\sqrt{}$ | (3, 3, 3, 3) | 88.9% | 53.8% | 90.2% | 84.9% | 71.7 | 105.5 | 39.1 |
| $\sqrt{}$ | (5, 5, 5, 5) | 91.4% | 56.6% | 93.8% | 84.2% | 71.7 | 105.5 | 39.1 |
| $\sqrt{}$ | (7, 7, 7, 7) | 91.4% | 55.8% | 95.2% | 82.7% | 71.7 | 105.5 | 39.7 |

Analysis of Table 3 yields the following insights:

1.  Adding the channel attention mechanism before the ELAN module significantly improved detection accuracy, with AP(0.5) and AP(0.5:0.95) increasing by 1.6% and 1%, respectively. The model's speed also increased by 0.6 FPS, although model size grew by 3.1 MB and complexity rose by 0.4 GB. This enhancement justifies the inclusion of channel attention before the ELAN module.
2.  Introducing a spatial attention mechanism with a $5 \times 5$ kernel size also improved detection speed, with a 3.5 MB reduction in model size and a 2.6 FPS speed-up. There was a slight increase in AP(0.5) (1.3%) but a minor decrease in AP(0.5:0.95) (0.6%), indicating a trade-off between speed and bounding box precision.
3.  Combining both channel and spatial attention mechanisms before the ELAN module achieved the highest overall accuracy, with a 56.6% AP(0.5:0.95) and a 92.2% AP(0.5) when using kernel sizes of [5, 7, 5, 3]. Despite no change in model size and complexity, detection speed decreased by 0.6 to 1.8 FPS compared to using only channel attention.

Based on these findings, we conclude that integrating both attention mechanisms before the ELAN module, despite a slight speed reduction, enhances detection accuracy. Therefore, we have adopted this approach, with a spatial attention mechanism using a $5 \times 5$ kernel size, which provided the best balance between accuracy metrics and model efficiency.

### 4.2.3. Ablation Experiment on EH-NMS

The effectiveness of the EH-NMS algorithm hinges on the area ratio threshold θ. A high θ can lead to the exclusion of valid bounding boxes, reducing recall, while a low θ may not sufficiently eliminate overlaps, especially in extreme cases. To find the best θ, ablation experiments on the AIRD dataset tested values from 1.0 to 1.5 with 0.05 increments. The upper limit ensures no correct boxes are wrongly dismissed, and the lower limit reflects the minimum intersection area ratio.

Tables 4 and 5 show EH-NMS's performance metrics for various θ values on SADD and SSDD [58]. Lower θ values improve recall and AP(0.5) and AP(0.5:0.95) but slightly lower precision. Higher θ values increase precision but decrease recall. Notably, θ values of 1.2 and 1.05 yield the highest AP(0.5) and AP(0.5:0.95) on SADD and SSDD, respectively, indicating that the optimal θ depends on the target geometry. Given this, EH-NMS performs strongly for SAR aircraft and ship detection within the 1.05 to 1.2 range, leading us to select θ = 1.2 for SAR aircraft detection.

**Table 4.** Ablation study of threshold θ on the SADD.

| θ | AP(0.5) | AP(0.5:0.95) | Precision | Recall |
|------|---------|--------------|-----------|--------|
| 1.00 | 89.8% | 55.3% | 92.6% | 82.7% |
| 1.05 | 89.8% | 55.3% | 93.0% | 82.7% |
| 1.10 | 89.9% | 55.4% | 93.0% | 82.7% |
| 1.15 | 90.0% | 55.4% | 93.0% | 82.7% |
| 1.20 | 90.1% | 55.4% | 93.0% | 82.7% |
| 1.25 | 89.5% | 55.0% | 92.9% | 81.6% |
| 1.30 | 88.1% | 54.3% | 92.8% | 80.5% |
| 1.35 | 86.9% | 53.7% | 92.8% | 80.5% |
| 1.40 | 87.0% | 53.8% | 92.8% | 80.5% |
| 1.45 | 86.9% | 53.6% | 90.1% | 83.5% |
| 1.50 | 86.3% | 53.2% | 90.7% | 82.4% |

**Table 5.** Ablation study of threshold θ on the SSDD.

| θ | AP(0.5) | AP(0.5:0.95) | Precision | Recall |
|------|---------|--------------|-----------|--------|
| 1.00 | 93.4% | 61.4% | 92.5% | 86.4% |
| 1.05 | 93.9% | 61.6% | 93.6% | 86.3% |
| 1.10 | 93.4% | 61.4% | 93.1% | 87.7% |
| 1.15 | 93.1% | 61.2% | 93.8% | 86.6% |
| 1.20 | 91.8% | 60.7% | 92.2% | 88.5% |
| 1.25 | 90.2% | 59.9% | 93.0% | 87.0% |
| 1.30 | 87.1% | 58.4% | 93.3% | 83.9% |
| 1.35 | 84.2% | 56.9% | 92.4% | 80.6% |
| 1.40 | 81.5% | 55.2% | 93.8% | 77.8% |
| 1.45 | 79.7% | 54.1% | 93.3% | 76.9% |
| 1.50 | 77.9% | 52.7% | 91.6% | 75.6% |

1. On both SADD and SSDD, AP(0.5) and AP(0.5:0.95) initially rise with increasing θ, then decline. This is due to EH-NMS effectively reducing overlaps at low θ but becoming overly restrictive at high θ, leading to the exclusion of valid bounding boxes and reduced accuracy.

2. For SADD, AP(0.5) and AP(0.5:0.95) peak at 90.1% and 55.4% with θ at 1.2. On SSDD, the highest values of 93.9% and 61.6% are achieved with θ at 1.05. This indicates that the optimal θ varies with target geometry, with 1.05 to 1.2 being effective for SAR aircraft and 1.00 to 1.10 for SAR ships.

3. Precision and recall show a dual fluctuation on both datasets, with initial increases followed by decreases at different stages. The first increase in precision is due to reduced false positives, while the second is from further reduction in overlaps. Recall initially rises as confident false positives are removed, but then falls as true positives are mistakenly excluded.

Our analysis confirms EH-NMS's superior performance in SAR target detection accuracy over classical NMS methods. Considering the metrics, a threshold θ of 1.2 is optimal for SAR aircraft detection. In comparison with Standard NMS, Soft-NMS [42], DIoU-NMS [44], and WBF on SADD, EH-NMS demonstrates its advantages, as detailed in Table 6.

Analysis of Table 6 yields the following insights:

1. EH-NMS excels in SAR aircraft detection by effectively eliminating highly overlapping bounding boxes, achieving superior results with 90.1% AP(0.5), 55.4% AP(0.5:0.95), and 93.0% precision. These figures surpass the next best techniques by 0.3%, 0.1%, and 0.2%, respectively.

2. Soft-NMS and WBF show a decline in accuracy, likely due to their optimization for natural images rather than SAR remote sensing. Meanwhile, DIoU-NMS demonstrates a slight enhancement in accuracy, indicating its robustness and generalizability.

To verify the robustness and generalizability of the EH-NMS, a comparative analysis was conducted between YOLOv7 [54], YOLOR-P6, and the proposed YOLO-SAD on the SADD, with and without the implementation of EH-NMS. Table 7 presents an overview of the key accuracy metrics, while Figure 9 displays the visualized results.

**Table 6.** Comparison of multiple NMS techniques applied to the SADD baseline, utilizing an IoU threshold of 0.6.

| Algorithm | AP(0.5) | AP(0.5:0.95) | Precision | Recall |
|---|---|---|---|---|
| NMS | 89.8% | 55.2% | 92.6% | 82.7% |
| Soft-NMS | 85.5% | 53.3% | 92.6% | 82.7% |
| DIoU-NMS | 89.9% | 55.3% | 92.6% | 82.7% |
| WBF | 88.4% | 43.5% | 92.8% | 80.0% |
| EH-NMS | 90.1% | 55.4% | 93.0% | 82.7% |

**Table 7.** Ablation study on EH-NMS, benchmark testing all models under the same computational environment. $\sqrt{}$ denotes the usage of EH-NMS; - indicates the usage of classical NMS.

| Model | NMS | AP(0.5) | AP(0.5:0.95) | Precision | Recall |
|---|---|---|---|---|---|
| YOLOv7 | - | 89.8% | 55.2% | 92.6% | 82.7% |
|  | $\sqrt{}$ | 90.1% | 55.4% | 93.0% | 82.7% |
| YOLOr-p6 | - | 89.2% | 53.5% | 93.6% | 80.5% |
|  | $\sqrt{}$ | 89.7% | 53.8% | 94.4% | 80.5% |
| our | - | 91.8% | 57.0% | 94.0% | 81.2% |
|  | $\sqrt{}$ | 91.9% | 57.1% | 94.0% | 81.2% |

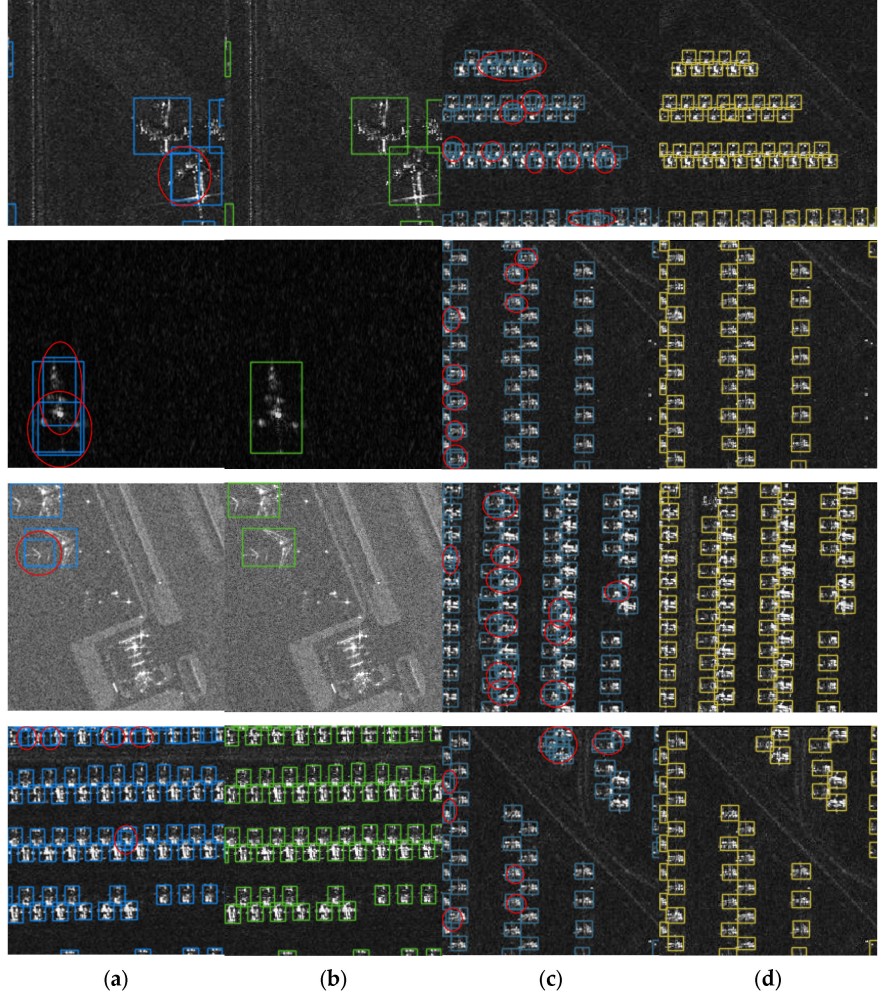

(a)  (b)  (c)  (d)

**Figure 9.** Comparison of detection results between EH-NMS and NMS: (**a**) YOLOv7; (**b**) YOLOv7 + EH-NMS; (**c**) YOLOr-p6; and (**d**) YOLOr-p6 + EH-NMS. The red lines delineate the erroneous overlapping detection boxes. The rectangles represent predicted bounding boxes; The red circles indicate these overlapped predicted bounding boxes.

Based on Table 7 and Figure 9, the following conclusions can be drawn:

1.  When applying the EH-NMS post-processing method to the YOLOv7 and YOLOr-p6 models demonstrates a significant improvement in the precision metric. Comparing with YOLOv7, there is a 0.4% improvement, while compared with YOLOr-p6, the improvement reaches 0.8%. Additionally, there is a slight improvement in the AP(0.5) metric, with an increase of 0.3% compared with YOLOv7 and 0.5% compared with YOLOr-p6. Similarly, there is a slight improvement in the AP(0.5:0.95) metric, with an increase of 0.2% compared with YOLOv7 and 0.3% compared with YOLOr-p6. Notably, the EH-NMS post-processing method does not result in a decrease in the recall metric.

2.  Through the visualized results in Figure 9, it is evident that the EH-NMS post-processing method accurately eliminates incorrectly overlapped bounding boxes, consequently resulting in improved visual outcomes. This experiment confirms the effectiveness of the proposed EH-NMS post-processing method.

### 4.2.4. Ablation Experiment on the Overall Network Structure

To substantiate the effectiveness of the proposed YOLO-SAD method, we conducted a thorough ablation study on the SADD, focusing on the three enhancement strategies. The results, summarized in Table 8, lead to the following conclusions:

**Table 8.** Ablation study on overall network structure. $\sqrt{}$ indicates the utilization of this module or strategy; - denotes the non-utilization of this module or strategy.

| Method | | | AP(0.5) | AP(0.5:0.95) | Precision | Recall | Params (M) | Flops (G) | FPS |
|---|---|---|---|---|---|---|---|---|---|
| SAD-FPN | A-ELAN-H | EH-NMS | | | | | | | |
| - | - | - | 89.8% | 55.2% | 92.6% | 82.7% | 74.8 | 105.1 | 40.3 |
| $\sqrt{}$ | - | - | 92.0% | 56.3% | 94.2% | 83.3% | 69.9 | 97.2 | 43.7 |
| - | $\sqrt{}$ | - | 91.4% | 56.6% | 93.8% | 84.2% | 71.7 | 105.5 | 39.1 |
| - | - | $\sqrt{}$ | 90.1% | 55.4% | 93.0% | 82.7% | 74.8 | 105.1 | 40.3 |
| $\sqrt{}$ | $\sqrt{}$ | - | 91.8% | 57.0% | 94.0% | 81.2% | 70.0 | 97.2 | 43.7 |
| - | $\sqrt{}$ | $\sqrt{}$ | 91.4% | 56.6% | 94.6% | 84.2% | 71.7 | 105.5 | 39.1 |
| $\sqrt{}$ | - | $\sqrt{}$ | 92.3% | 56.4% | 94.2% | 83.3% | 69.9 | 97.2 | 43.7 |
| $\sqrt{}$ | $\sqrt{}$ | $\sqrt{}$ | 91.9% | 57.1% | 94.0% | 81.2% | 70.0 | 97.2 | 43.7 |

1.  The integration of the lightweight SAD-FPN architecture into the network resulted in a 3.4 FPS increase and improvements in AP(0.5) and AP(0.5:0.95) of 2.2% and 1.1%, respectively, over the baseline. Despite a 1.2 FPS decrease upon adding the A-ELAN-H module, detection accuracy saw a significant boost, with AP(0.5) and AP(0.5:0.95) improving by 1.6% and 1.4%, respectively.

2.  Combining SAD-FPN with the A-ELAN-H module led to a 0.7% increase in AP(0.5:0.95) and a 2% rise in AP(0.5) over using SAD-FPN alone. Compared to the baseline, this configuration achieved a 91.8% AP(0.5) and a 57.0% AP(0.5:0.95). The SAD-FPN structure's efficiency and A-ELAN-H's accuracy complement each other, reducing the additional complexity introduced by A-ELAN-H. This combination decreased the model size by 1.7 MB and complexity by 8.3 GB compared to using A-ELAN-H alone, and improved detection speed by 4.6 FPS.

3.  The EH-NMS post-processing method slightly improves AP(0.5) and AP(0.5:0.95) without affecting model size, complexity, or speed. When applied to the SAD-FPN and A-ELAN-H combination, it further increased AP(0.5:0.95) to 57.1% and AP(0.5) to 91.9%.

In conclusion, the fusion of SAD-FPN and A-ELAN-H in YOLO-SAD not only leverages their individual strengths but also mitigates the speed limitations of A-ELAN-H. This integration delivers exceptional accuracy and speed, making it the optimal configura-

tion for YOLO-SAD. Additionally, the use of EH-NMS further refines detection accuracy, highlighting its effectiveness in enhancing YOLO-SAD's performance.

### 4.3. Comparative Experiment

To validate the effectiveness of our proposed object detection method, we conducted comparative experiments with five state-of-the-art (SOTA) networks, which were retrained on the SADD dataset. Figure 10 illustrates the visual outcomes of these methods on SADD, while Table 9 presents the key performance metrics. These results allow us to conclude the following:

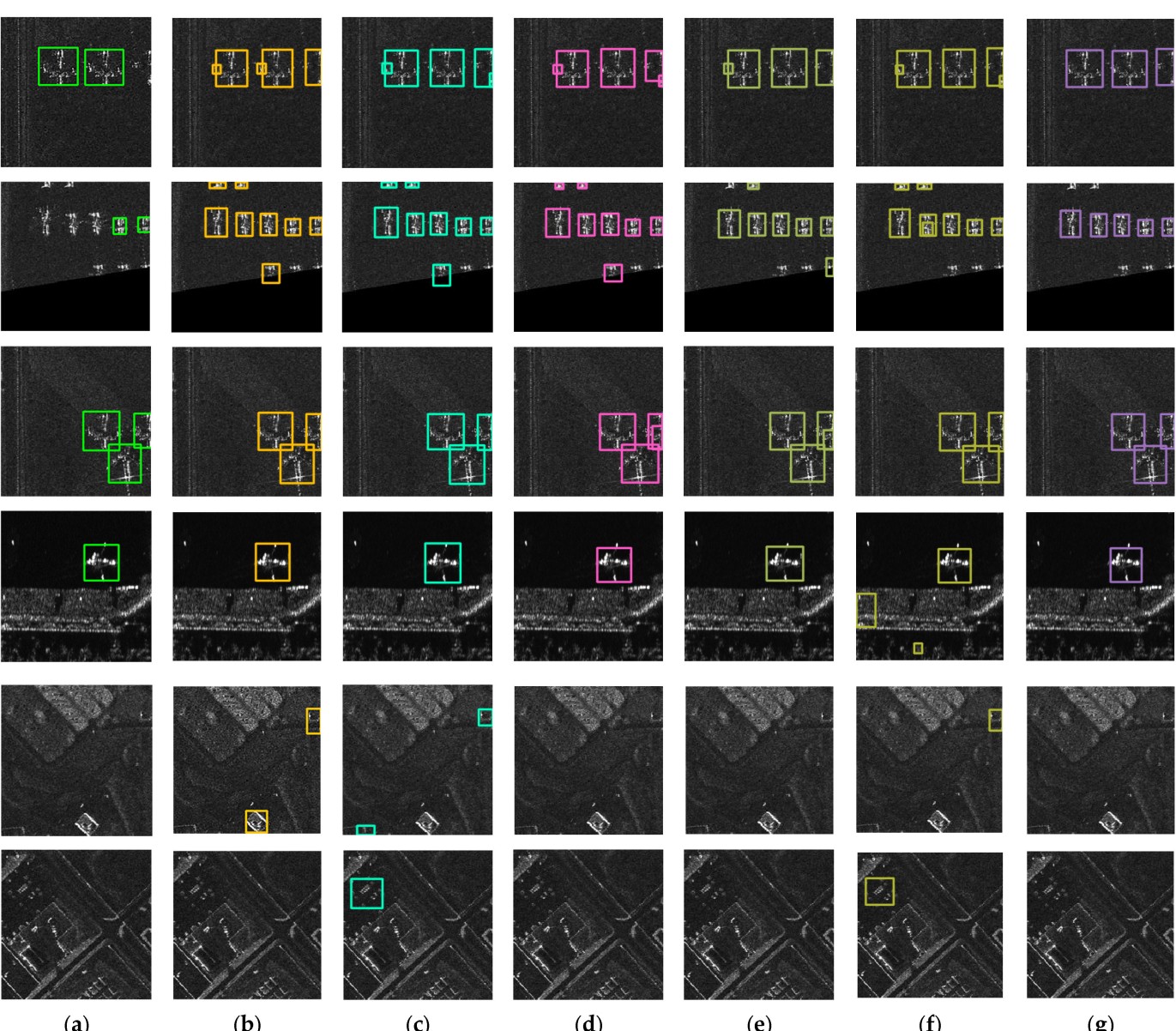

**Figure 10.** Visualization of different methods on the SADD: (**a**) ground truth; (**b**) ResNet50-CSP; (**c**) ResNeXt50-CSP; (**d**) YOLOv4; (**e**)YOLOr-p6; (**f**) YOLOv7; and (**g**) YOLO-SAD. The green rectangles represent the true labels, while the rectangles of other colors correspond to the predicted bounding boxes from different methods.

1.  Our YOLO-SAD network outperforms the baseline on SADD, with a 2.1% increase in AP(0.5) to 91.9% and a 1.9% increase in AP(0.5:0.95) to 57.1%. Detection speed also slightly improved from 41.6 to 43.7 FPS. The SAD-FPN structure efficiently reduces high-level feature redundancy in PAN, and the attention mechanisms do not significantly increase model complexity. YOLO-SAD thus balances accuracy and speed effectively.
2.  YOLO-SAD exceeds other SOTA methods on SADD. ResNeXt50-CSP, the next best, achieves 91.5% AP(0.5) and 55.9% AP(0.5:0.95), which are lower than YOLO-SAD's 91.9% and 57.1%. This indicates YOLO-SAD's high accuracy in SAR aircraft detection.
3.  YOLO-SAD's detection speed is 43.7 FPS on SADD, a 2 FPS improvement over the baseline and 2.8 FPS over YOLO-V4. Although it does not match the highest speed networks like ResNet50-CSP, YOLOr-p6, and ResNeXt50-CSP, its accuracy is notably superior. YOLO-SAD's 0.5% improvement in AP(0.5) and 1.2% in AP(0.5:0.95) over the top-performing ResNeXt50-SCP justifies the slight speed difference.

**Table 9.** Comparative experiment with SOTA fast object detectors on the SADD. The conducted benchmark testing of all models under the same computational environment.

| Model | Epochs | AP(0.5) | AP(0.5:0.95) | Precision | Recall | Params (M) | Flops (G) | FPS |
|---|---|---|---|---|---|---|---|---|
| ResNet50-CSP | 300 | 91.5% | 55.9% | 92.3% | 83.5% | 65.1 | 64.4 | 55.7 |
| ResNeXt50-CSP | 300 | 91.3% | 55.4% | 88.9% | 85.3% | 64.3 | 59.7 | 56.9 |
| YOLOv4 | 300 | 87.8% | 54.0% | 93.8% | 77.5% | 105.5 | 119.7 | 40.9 |
| YOLOr-p6 | 300 | 89.2% | 53.5% | 93.6% | 80.5% | 74.3 | 80.6 | 56.9 |
| YOLOv7 | 300 | 89.8% | 55.2% | 92.6% | 82.7% | 74.8 | 105.1 | 41.6 |
| Our | 300 | 91.9% | 57.1% | 94.0% | 81.2% | 69.9 | 97.2 | 43.7 |

To further evaluate YOLO-SAD's generalization, we compared it with four leading networks on the SAR-AIRcraft-1.0 dataset [55]. Visual results are shown in Figure 11, and performance metrics are detailed in Table 10, all under the same computational conditions.

**Table 10.** Comparative experiment with SOTA fast object detectors on the SAR-AIRcraft-1.0. The conducted benchmark testing of all models under the same computational environment.

| Model | Epochs | AP(0.5) | AP(0.5:0.95) | Precision | Recall | Params (M) | Flops (G) | FPS |
|---|---|---|---|---|---|---|---|---|
| ResNet50-CSP | 400 | 88.9% | 60.9% | 89.5% | 84.2% | 65.1 | 64.4 | 55.7 |
| ResNeXt50-CSP | 400 | 82.9% | 51.5% | 84.1% | 77.8% | 64.3 | 59.7 | 56.9 |
| YOLOv4 | 400 | 81.6% | 51.3% | 85.9% | 73.3% | 105.5 | 119.7 | 40.9 |
| YOLOr-p6 | 400 | 80.8% | 51.6% | 81.4% | 77.2% | 74.3 | 80.6 | 56.9 |
| YOLOv7 | 400 | 89.8% | 63.0% | 88.0% | 87.8% | 74.8 | 105.1 | 41.6 |
| Our | 400 | 90.8% | 63.8% | 89.3% | 87.9% | 69.9 | 97.2 | 43.7 |

Figure 11 and Table 10 confirm the effectiveness of YOLO-SAD, echoing the SADD findings. YOLO-SAD delivered the highest detection accuracy at 90.8% AP(0.5), 63.8% AP(0.5:0.95), and 87.9% recall, outperforming the next best method by margins of 1.0%, 0.8%, and 0.1% in each metric. YOLO-SAD also maintained superior detection speed, consistent with its performance on SADD. These achievements across the 2966 varied SADD images and the 4368 differently sized SAR-AIRcraft-1.0 images underscore the model's robust generalization ability.

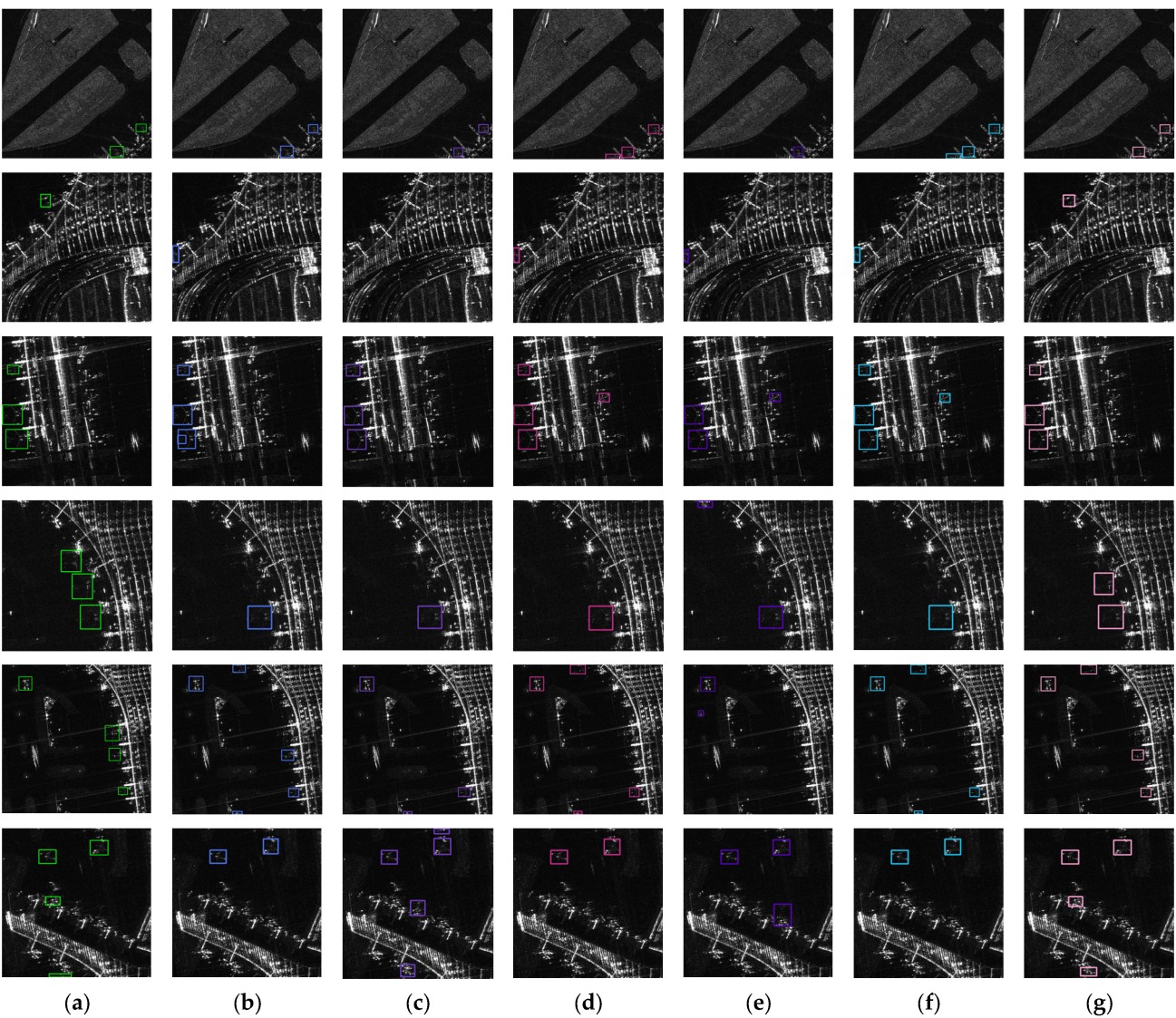

**Figure 11.** Performance of different methods on the SAR-AIRcraft-1.0 dataset: (**a**) ground truth; (**b**) ResNet50-CSP; (**c**) ResNeXt50-CSP; (**d**) YOLOv4; (**e**)YOLOr-p6; (**f**) YOLOv7; and (**g**) YOLO-SAD. The green rectangles denote the ground true labels, while the rectangles of other colors depict the predicted bounding boxes generated by different methods.

## 5. Discussion

This paper introduces optimized strategies for SAR aircraft detection, rigorously tested through ablation and comparative experiments. The SAD-FPN structure outperforms traditional PAN by reducing parameters and simplifying the model, leading to improved speed and accuracy. Attention mechanisms in the neck's feature encoding module further boost accuracy without a significant increase in complexity. The YOLO-SAD network, integrating these enhancements, surpasses YOLOv7 in both speed and accuracy.

Comparative experiments revealed that less parameter-heavy networks like ResNet50-CSP and ResNeXt50-CSP achieve high accuracy on SADD, likely due to less overfitting on this small-scale dataset.

Our YOLO-SAD with EH-NMS post-processing significantly reduces false positives compared to other SOTA methods, enhancing accuracy and reliability—vital for creating SAR aircraft datasets. Given the limited availability of high-quality public SAR datasets, there is a pressing need for a large-scale, exceptional quality dataset. We aim to apply our strategy to meet this demand.

Result visualization during experiments also highlighted issues with SADD's ground truth labels. For example, an aircraft at the image edge was omitted from the labels, possibly due to the slicing process. Such inaccuracies can mislead the neural network and affect performance metrics. Despite this, our method successfully detected the overlooked aircraft, demonstrating its robustness and potential for dataset generation and enhancement.

## 6. Conclusions

This paper introduces a specialized object detection network for the swift and precise detection of aircraft in SAR images. To optimize speed and accuracy, we implement three key enhancements: the SAR-optimized Feature Pyramid Network (SAD-FPN) which fuses features through addition; the Attention-Efficient Layer Aggregation Network-Head (A-ELAN-H) module that emphasizes critical features using attention mechanisms; and the Enhanced Non-Maximum Suppression (EH-NMS) algorithm that refines detection results by eliminating overlapping boxes.

Ablation studies confirm the significant impact of these improvements on detection accuracy. Our method outperforms five state-of-the-art networks on the SADD dataset in terms of AP(0.5) and AP(0.5:0.95), while maintaining a balance between computational efficiency and performance. The effectiveness of SAD-FPN in reducing overfitting and managing redundant high-level features is crucial, especially for small datasets like SADD. The A-ELAN-H module enhances feature utilization by focusing on essential channels and spatial locations.

Looking ahead, our research will focus on the following areas:

1. Developing larger-scale SAR aircraft detection datasets for fine-grained classification, enabling the exploration of network architectures tailored to SAR features.
2. Investigating neural network design strategies for small-scale datasets, exploring data augmentation, meta-learning, self-supervised learning, and other techniques suited for SAR object detection.

**Author Contributions:** Conceptualization, J.C. and Q.Z.; methodology, J.C.; writing the original draft, J.C. and Y.S.; formal analysis, Q.Z.; visualization, Y.L.; writing, review, editing, and supervision J.C., Z.W., Y.S. and Q.Z. All authors have read and agreed to the published version of the manuscript.

**Funding:** This research was funded by the Shenzhen Science and Technology Program (No. ZDSYS202106623091808026), and the National Key Research and Development Program of China (No. 2022YFE0209300).

**Institutional Review Board Statement:** Not applicable.

**Informed Consent Statement:** Not applicable.

**Data Availability Statement:** Publicly available datasets were analyzed in this study. The SADD introduced in this study is openly accessible and can be received through [3]. The SAR-AIRcraft-1.0 dataset introduced in this study is openly accessible and can be received through [55].

**Acknowledgments:** The authors would like to appreciate Kin-Yiu Wong sharing the codes of YOLOV7 at: GitHub—WongKinYiu/yolov7: Implementation of paper—YOLOv7: Trainable bag-of-freebies sets new state-of-the-art for real-time object detectors.

**Conflicts of Interest:** The authors declare no conflicts of interest.

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
