# Peer review of "YOLO-SAD: An Efficient SAR Aircraft Detection Network"

_applsci, doi:10.3390/app14073025_

Round 1

Reviewer 1 Report

Comments and Suggestions for Authors

The Manuscript presents a novel approach called YOLO-SAD for aircraft detection in SAR images of airports. It highlights the challenges faced by existing methods, such as low resolution, complex scenes, accuracy issues, overlapping detections, and missed targets. YOLO-SAD addresses these challenges by leveraging the Attention-Efficient Layer Aggregation Network-Head (A-ELAN-H) module for feature prioritization, the SAR Aircraft Detection-Feature Pyramid Network (SAD-FPN) for multi-scale feature fusion optimization, and Enhanced Non-Maximum Suppression (EH-NMS) for eliminating overlapping detections. The proposed method achieves promising results on the SAR Aircraft Detection Dataset (SADD), surpassing the baseline and outperforming five state-of-the-art methods in terms of average precision (AP) metrics.

However, some potential weaknesses of the manuscript could include:

The manuscript lacks information on the reproducibility of the results, such as details about the implementation, hyperparameters, and code availability, which are essential for other researchers to validate and build upon the proposed method.

While the manuscript discusses various feature fusion methods, such as feature concatenation, feature addition, and feature multiplication, it lacks a comprehensive comparison of these methods in terms of their impact on detection performance and computational efficiency. Further analysis and comparison could provide insights into the most suitable fusion method for SAR aircraft detection.

The manuscript describes the integration of attention mechanisms in the feature encoding module to enhance detection accuracy. However, it does not thoroughly explore the potential drawbacks or limitations of these mechanisms, such as increased computational complexity or potential overfitting issues. A more detailed discussion of the trade-offs associated with incorporating attention mechanisms would provide a clearer understanding of their impact on the proposed method.

The manuscript discusses the selection of the optimal threshold (θ) for the Enhanced Non-Maximum Suppression (EH-NMS) algorithm. While it presents results for different threshold values, it does not provide a comprehensive analysis of the sensitivity of the algorithm to threshold variations or the potential trade-offs between recall and precision. A more thorough investigation of the effects of threshold selection on detection performance could enhance the robustness of the proposed method.

The manuscript acknowledges limitations in the ground truth labels of the SAR Aircraft Detection Dataset (SADD), noting instances where aircraft are missed in the provided labels. However, it does not discuss the potential impact of these labeling inaccuracies on the evaluation of the proposed method or strategies for mitigating their effects. A more in-depth analysis of dataset limitations and their implications for evaluation would strengthen the validity of the experimental results.

While the manuscript demonstrates promising results on the SADD, it lacks a discussion of the generalization capabilities of the proposed method to unseen datasets or real-world scenarios. Further evaluation on diverse datasets and under different environmental conditions would provide a more comprehensive assessment of the practical applicability of the proposed method.

Comments on the Quality of English Language

Minor

Author Response

Point-to-point responses to reviewer’s comments on manuscript

applsci-2881145

Titled “YOLO-SAD: An Efficient SAR Aircraft Detection Network”

  We would like to take this opportunity to thank you for your helpful and constructive comments about our manuscript. The comments and our point-by-point responses are listed as the following:

  1. The manuscript lacks information on the reproducibility of the results, such as details about the implementation, hyperparameters, and code availability, which are essential for other researchers to validate and build upon the proposed method.

A: Thank you for your valuable suggestion. We have taken it into account and made the following updates to our paper. Firstly, we have expanded Section 3 to include a comprehensive and detailed description of the proposed YOLO-SAD network. Our focus was to provide clear explanations and visual representations of the modules depicted in Fig 1, Fig 2, Fig 4, and Fig 5. This additional information will assist readers to better understand the architecture of our proposed network. Furthermore, we have addressed certain previously overlooked deployment details and key hyper-parameters in Section 4.1.2, which significantly contributes to the reproducibility of our approach. By including these crucial aspects, we believe that readers will have all the necessary information to validate and build upon our method. Lastly, we have included a clarification regarding the datasets availability in the Data Availability Statement.

  1. While the manuscript discusses various feature fusion methods, such as feature concatenation, feature addition, and feature multiplication, it lacks a comprehensive comparison of these methods in terms of their impact on detection performance and computational efficiency. Further analysis and comparison could provide insights into the most suitable fusion method for SAR aircraft detection.

A: Thank you for this valuable suggestion. We have taken it into account and made the following updates to our paper in Section 4.2.1. We have included a comprehensive comparison of the three feature fusion methods, namely feature concatenation, indirect fusion, and feature addition, in relation to their impact on detection performance and computational efficiency. Additionally, we have delved into further discussions and provided detailed explanations regarding the selection and deployment of feature fusion methods.

  1. The manuscript describes the integration of attention mechanisms in the feature encoding module to enhance detection accuracy. However, it does not thoroughly explore the potential drawbacks or limitations of these mechanisms, such as increased computational complexity or potential overfitting issues. A more detailed discussion of the trade-offs associated with incorporating attention mechanisms would provide a clearer understanding of their impact on the proposed method.

A: Thank you for your valuable suggestion. We have taken it into account and made the following updates to our paper in Section 4.2.2. We have included a comprehensive comparison between spatial attention mechanisms and channel attention mechanisms, focusing on their impact on detection performance and computational efficiency. Furthermore, we conducted an in-depth analysis of the strengths and limitations associated with these two attention mechanisms. Additionally, we made further discussions and provided explanations regarding the selection and deployment of attention mechanisms.

  1. The manuscript discusses the selection of the optimal threshold (θ) for the Enhanced Non-Maximum Suppression (EH-NMS) algorithm. While it presents results for different threshold values, it does not provide a comprehensive analysis of the sensitivity of the algorithm to threshold variations or the potential trade-offs between recall and precision. A more thorough investigation of the effects of threshold selection on detection performance could enhance the robustness of the proposed method.

A: Thank you for this valuable suggestion, which is really constructive. We carefully went through this section and made the following updates to our paper in Section 4.2.3. We have included an in-depth analysis of the correlation between thresholds () and key metrics such as AP(0.5), AP(0.5:0.95), Precision, and Recall. In this analysis, we elucidated the impact of  on these metrics and made further discussions and explanations on the process of striking a balance and selecting the optimal  among these metrics.

  1. The manuscript acknowledges limitations in the ground truth labels of the SAR Aircraft Detection Dataset (SADD), noting instances where aircraft are missed in the provided labels. However, it does not discuss the potential impact of these labeling inaccuracies on the evaluation of the proposed method or strategies for mitigating their effects. A more in-depth analysis of dataset limitations and their implications for evaluation would strengthen the validity of the experimental results.

A: Thank you for your valuable suggestion. We have taken it into account and made the following updates to our paper in the third paragraph of the Discussion section. We have included further discussions about the limitations of the dataset. In this discussion, we thoroughly analyzed the underlying factors behind these limitations and assessed their impact on both network training and the evaluation of detection accuracy.

  1. While the manuscript demonstrates promising results on the SADD, it lacks a discussion of the generalization capabilities of the proposed method to unseen datasets or real-world scenarios. Further evaluation on diverse datasets and under different environmental conditions would provide a more comprehensive assessment of the practical applicability of the proposed method.

A: Thank you for your valuable suggestion. We have taken it into account and made the following updates to our paper. In Section 4.3, we have included a comparative experiment on a second dataset to further validate the generalization capability of YOLO-SAD. This experiment involved comparing YOLO-SAD with five other state-of-the-art object detection networks on a subset of the MSAR dataset, which consists of sliced images of airplane targets. Figure 9 presents the visual results obtained by different methods on this dataset, while Table 9 showcases the key performance metrics.

By analyzing Figure 9 and Table 9, we arrived at conclusions similar to those obtained from the SADD dataset. YOLO-SAD demonstrated the highest detection accuracy, achieving 80.3% AP(0.5), 28.3% AP(0.5:0.95), 83.0% Precision. Compared with the second-ranked method, YOLO-SAD exhibited improvements of 0.6% in AP(0.5), 0.8% in AP(0.5:0.95), 5.4% in Precision. Visual analysis revealed that YOLO-SAD had fewer false positive results compared with other methods, while also demonstrating superior small target detection capabilities in low-quality SAR remote sensing images. In terms of detection speed, the experimental results were consistent with those obtained on the SADD dataset.

The outcomes of this comparative experiment aligned with the previous comparative experiments conducted on the SADD dataset, thus confirming the robust generalization capability of YOLO-SAD.

Reviewer 2 Report

Comments and Suggestions for Authors

This paper, “YOLO-SAD: An Efficient SAR Aircraft Detection Network”.

In my opinion this paper addresses a topic of great importance nowadays. The results obtained seem to be very promising.

This paper is well written and structured, it is very easily understandable, and the results   obtained are very interesting. However, there are some details that are not clear.

I would like to know the details of the algorithm YOLO-SAD, although they are described in the article, they are not enough to reproduce the proposed method.

Author Response

Point-to-point response to reviewer’s comment on manuscript

applsci-2881145

Titled “YOLO-SAD: An Efficient SAR Aircraft Detection Network”

  We would like to take this opportunity to thank you for this helpful and constructive comment about our manuscript. The comment and our point-by-point response is listed as the following:

  1. I would like to know the details of the algorithm YOLO-SAD, although they are described in the article, they are not enough to reproduce the proposed method.

A: Thank you for your valuable suggestion. We have taken it into account and made the following updates to our paper. Firstly, we have expanded Section 3 to include a comprehensive and detailed description of the proposed YOLO-SAD network. Our focus was to provide clear explanations and visual representations of the modules depicted in Fig 1, Fig 2, Fig 4, and Fig 5. This additional information will assist readers to better understand the architecture of our proposed network. Furthermore, we have addressed certain previously overlooked deployment details and key hyper-parameters in Section 4.1.2, which significantly contributes to the reproducibility of our approach. By including these crucial aspects, we believe that readers will have all the necessary information to validate and build upon our method.

Reviewer 3 Report

Comments and Suggestions for Authors

The manuscript applsci-2881145 presents a DL algorithm for detection of aircrafts in SAR images. The study is very well documented with recent references relevant to the topic. The algorithm proposed in this paper was described thoroughly allowing full reproduction of the study.

It is recommended to add a short discussion on the processing spped, which could be an important performance metric for such detection problems. 

Author Response

Point-to-point response to reviewer's comment on manuscript

applsci-2881145

Titled “YOLO-SAD: An Efficient SAR Aircraft Detection Network”

  We would like to take this opportunity to thank you for this helpful and constructive comment about our manuscript. The comment and our point-by-point response is listed as the following:

  1. It is recommended to add a short discussion on the processing speed, which could be an important performance metric for such detection problems. 

A: Thank you for your valuable suggestion. We have taken it into consideration and made the following updates to our paper. In Section 4.2.1, we have included detailed discussions on the detection speed of various feature fusion methods employed in the SAD-FPN. Furthermore, in Section 4.2.2, we have provided in-depth discussions on the detection speed in relation to different attention mechanisms implemented in the A-ELAN-H module. Additionally, in Section 4.3, during the analysis of the comparative experimental results, we have included a comprehensive comparison and analysis of the detection speed between YOLO-SAD and five other state-of-the-art object detection networks.

Reviewer 4 Report

Comments and Suggestions for Authors

1. How was the performance of the proposed YOLO-SAD method compared to other state-of-the-art object detection networks on the SADD?

2. What were the primary conclusions drawn from the comparative analysis of different detection methods?

3. How did the proposed YOLO-SAD method demonstrate a balance between accuracy and computational efficiency?

Author Response

Point-to-point responses to reviewer’s comments on manuscript

applsci-2881145

Titled “YOLO-SAD: An Efficient SAR Aircraft Detection Network”

  We would like to take this opportunity to thank you for your helpful and constructive comments about our manuscript. The comments and our point-by-point responses are listed as the following:

  1. How was the performance of the proposed YOLO-SAD method compared to other state-of-the-art object detection networks on the SADD?

A: The comparative experiments conducted on the SADD dataset demonstrate that YOLO-SAD surpasses the baseline in terms of AP(0.5), AP(0.5:0.95), Precision, and FPS, with improvements of 2.1%, 1.9%, 1.4%, and 2.1 respectively. When compared with other SOTA object detection networks, YOLO-SAD still achieves the highest scores in AP(0.5), AP(0.5:0.95), and Precision, which are 91.9%, 57.1%, and 94% correspondingly. These results represent an improvement of 0.4%, 1.2%, and 0.2% respectively, when compared with the network ranked second.

  1. What were the primary conclusions drawn from the comparative analysis of different detection methods?

A: First and foremost, YOLO-SAD achieves the highest scores in terms of detection accuracy, specifically AP(0.5), AP(0.5:0.95), and Precision. This outcome substantiates the effectiveness of our proposed methodology, which involves reducing the influence of high-level features in feature fusion and incorporating attention mechanisms. These enhancements notably improve the network's detection accuracy in SAR aircraft detection tasks.

Furthermore, in terms of computational efficiency, YOLO-SAD achieves an FPS of 43.7 on the SADD dataset. This represents an improvement of 2 FPS over the baseline and 2.8 FPS over YOLO-V4. While YOLO-SAD may not match the FPS performance of high-performance networks such as ResNet50-CSP, YOLOr-p6, and ResNeXt50-CSP, it demonstrates a substantial advantage in terms of detection accuracy. When compared with the network with the highest detection accuracy among those lightweight network, ResNeXt50-SCP, YOLO-SAD exhibits a 0.5% improvement in AP(0.5) and a 1.2% improvement in AP(0.5:0.95). Therefore, considering the considerable gains in detection accuracy, we deem the disparity in computational efficiency to be acceptable.

  1. How did the proposed YOLO-SAD method demonstrate a balance between accuracy and computational efficiency?

A: On one hand, the baseline we adopted is YOLOV7, which is renowned for its balance between detection accuracy and computational efficiency. When compared with YOLOV5, YOLOV7 demonstrates a remarkable 120% increase in FPS while maintaining the same level of detection accuracy. Additionally, at the same detection speed, YOLOV7 exhibits a 0.5% improvement in detection accuracy.

One the other hand, we extensively discuss this aspect in the ablation experiments conducted in Section 4.2.4 and the comparative experiments outlined in Section 4.3. These sections emphasize the effectiveness of our proposed SAD-FPN structure, which effectively enhances both detection accuracy and computational efficiency. Furthermore, the A-ELAN-H module contributes to improved detection accuracy, albeit with a slight reduction in computational efficiency. By combining these two approaches, we can further enhance detection accuracy while compensating for any impact on computational efficiency caused by the A-ELAN-H module.

Round 2

Reviewer 1 Report

Comments and Suggestions for Authors

The reviewer is unconvinced by the authors' explanation and doubts the results. Therefore, the editor should make the final decision.

Comments on the Quality of English Language

Since there is a 20% similarity in the text, it requires rewriting.

Author Response

Point-to-point responses to reviewer’s comments on manuscript

applsci-2881145

Titled “YOLO-SAD: An Efficient SAR Aircraft Detection Network”

    We would like to take this opportunity to thank you for your helpful and constructive comments about our manuscript. The comments and our point-by-point responses are listed as the following:

  1. The reviewer is unconvinced by the authors' explanation and doubts the results. Therefore, the editor should make the final decision.

A: Thank you for this valuable suggestion. We thought your doubts may be attributed to the limitation of conducting additional comparative experiments with the subset of the MSAR dataset, since it is relatively small size and low resolution. Therefore, the substituted experiment is carried on as follow:

    To substantiate the robustness of our method, we supplemented our analysis with a comparative experiment using the SAR-AIRcraft-1.0 dataset. This comprehensive dataset, sourced from the Gaofen-3 satellite, features a 1m*1m spatial resolution and includes 4,368 image slices with 16,463 annotated instances across four sizes: 800*800, 1,000*1,000, 1,200*1,200, and 1,500*1,500 pixels. We segmented the dataset into an 80:20 split for training and validation, allocating 3,499 images with 13,812 instances to training and 869 images with 2,651 instances to validation.

    The additional experiment aimed to assess the generalization capability of YOLO-SAD. We benchmarked YOLO-SAD against four leading object detection networks on the SAR-AIRcraft-1.0 dataset. Figure 11 displays the visual outcomes of the various methods, while Table 10 details their performance metrics, all measured under identical computational conditions.

Table 10. Comparative Experiment with SOTA Fast Object Detectors on the SAR-AIRcraft-1.0. The conducted benchmark testing of all models under the same computational environment.

Model

Epochs

AP0.5

AP0.5:0.95

Precision

Recall

Params

(M)

Flops

(G)

FPS

ResNet50-CSP

400

88.9%

60.9%

89.5%

84.2%

65.1

64.4

55.7

ResNeXt50-CSP

400

82.9%

51.5%

84.1%

77.8%

64.3

59.7

56.9

YOLOv4

400

81.6%

51.3%

85.9%

73.3%

105.5

119.7

40.9

YOLOr-p6

400

80.8%

51.6%

81.4%

77.2%

74.3

80.6

56.9

YOLOv7

400

89.8%

63.0%

88.0%

87.8%

74.8

105.1

41.6

Our

400

90.8%

63.8%

89.3%

87.9%

69.9

97.2

43.7

Figure 11. Performance of different methods on the SAR-AIRcraft-1.0 dataset. (a) Ground truth. (b) ResNet50-CSP. (c) ResNeXt50-CSP. (d) YOLOv4. (e)YOLOr-p6. (f) YOLOv7. (g) YOLO-SAD. (Figure 11 is correctly exhibited in word file.)

    Upon reviewing Figure 11 and Table 10, our findings aligned with those from the SADD, reinforcing the effectiveness of YOLO-SAD. YOLO-SAD achieved the highest detection accuracy with 90.8% AP(0.5), 63.8% AP(0.5:0.95), and 87.9% Recall, surpassing the second-best method by 1.0%, 0.8%, and 0.1% in respective metrics. Consistent with SADD results, YOLO-SAD also maintained a leading position in detection speed. These results across diverse datasets—SADD with 2,966 images of varying spatial resolutions and SAR-AIRcraft-1.0 with 4,368 images of different sizes—demonstrate the model's strong generalization capabilities.

  1. Since there is a 20% similarity in the text, it requires rewriting.

A: Thank you for this valuable suggestion. we have rewritten a significant portion of the article to reduce similarity indices. Where it was challenging to avoid overlapping content, we have clearly indicated the sources and provided appropriate citations.

Reviewer 4 Report

Comments and Suggestions for Authors

1. The diversity and quality of the dataset are crucial for training robust models. How did you ensure the representativeness and adequacy of the SAR datasets used in your experiments?

2. Your proposed YOLO-SAD method integrates several novel components such as attention mechanisms and feature fusion techniques. Could you elaborate on the motivations behind each of these components and how they contribute to improving detection performance?

3. The Enhanced Hierarchical Non-Maximum Suppression (EH-NMS) algorithm seems to be a significant contribution to refining detection accuracy. Could you provide more details on how this algorithm works and its effectiveness compared to traditional NMS techniques?

Author Response

Point-to-point responses to reviewer’s comments on manuscript

applsci-2881145

Titled “YOLO-SAD: An Efficient SAR Aircraft Detection Network”

    We would like to take this opportunity to thank you for your helpful and constructive comments about our manuscript. The comments and our point-by-point responses are listed as the following:

  1. The diversity and quality of the dataset are crucial for training robust models. How did you ensure the representativeness and adequacy of the SAR datasets used in your experiments?

A: Thank you for your valuable suggestion. We have taken it into account and made the following updates to our paper. Due to the small scale of subset of MSAR dataset, it maybe unstable and unrepresentative, we conducted another comparative experiment in SAR-AIRcraft-1.0 dataset. The introduction of this dataset was added as follow:

    To substantiate the robustness of our method, we supplemented our analysis with a comparative experiment using the SAR-AIRcraft-1.0 dataset. This comprehensive dataset, sourced from the Gaofen-3 satellite, features a 1m*1m spatial resolution and includes 4,368 image slices with 16,463 annotated instances across four sizes: 800*800, 1,000*1,000, 1,200*1,200, and 1,500*1,500 pixels. We segmented the dataset into an 80:20 split for training and validation, allocating 3,499 images with 13,812 instances to training and 869 images with 2,651 instances to validation.

The additional comparative experiment result was added as follow:

    The additional experiment aimed to assess the generalization capability of YOLO-SAD. We benchmarked YOLO-SAD against four leading object detection networks on the SAR-AIRcraft-1.0 dataset. Figure 11 displays the visual outcomes of the various methods, while Table 10 details their performance metrics, all measured under identical computational conditions.

Table 10. Comparative Experiment with SOTA Fast Object Detectors on the SAR-AIRcraft-1.0. The conducted benchmark testing of all models under the same computational environment.

Model

Epochs

AP0.5

AP0.50.95

Precision

Recall

Params

(M)

Flops

(G)

FPS

ResNet50-CSP

400

88.9%

60.9%

89.5%

84.2%

65.1

64.4

55.7

ResNeXt50-CSP

400

82.9%

51.5%

84.1%

77.8%

64.3

59.7

56.9

YOLOv4

400

81.6%

51.3%

85.9%

73.3%

105.5

119.7

40.9

YOLOr-p6

400

80.8%

51.6%

81.4%

77.2%

74.3

80.6

56.9

YOLOv7

400

89.8%

63.0%

88.0%

87.8%

74.8

105.1

41.6

Our

400

90.8%

63.8%

89.3%

87.9%

69.9

97.2

43.7

Figure 11. Visualization of different methods on the SAR-AIRcraft-1.0 dataset. (a) Ground truth. (b) ResNet50-CSP. (c) ResNeXt50-CSP. (d) YOLOv4. (e)YOLOr-p6. (f) YOLOv7. (g) YOLO-SAD. (Figured 11 can correctly show in word file.)

    Upon reviewing Figure 11 and Table 10, our findings aligned with those from the SADD, reinforcing the effectiveness of YOLO-SAD. YOLO-SAD achieved the highest detection accuracy with 90.8% AP(0.5), 63.8% AP(0.5:0.95), and 87.9% Recall, surpassing the second-best method by 1.0%, 0.8%, and 0.1% in respective metrics. Consistent with SADD results, YOLO-SAD also maintained a leading position in detection speed. These results across diverse datasets—SADD with 2,966 images of varying spatial resolutions and SAR-AIRcraft-1.0 with 4,368 images of different sizes—demonstrate the model's strong generalization capabilities.

  1. Your proposed YOLO-SAD method integrates several novel components such as attention mechanisms and feature fusion techniques. Could you elaborate on the motivations behind each of these components and how they contribute to improving detection performance?

A: Thank you for your valuable suggestion. We have taken it into account and made the following updates to our paper. To elaborate on the motivations and explain how SAD-FPN contributes to improving detection performance, the following additions have been made:

    The traditional PAN architecture, depicted in Figure 3, integrates a top-down and bottom-up FPN. The top-down pathway enriches lower-level features with semantic information through up-sampling, while the bottom-up pathway refines higher-level features with spatial details through down-sampling. However, SAR images present unique challenges such as low resolution and coherent noise, which tend to accumulate in the backbone network and affect higher-level features. Relying heavily on these features during fusion can impair detection performance for SAR images.

Figure 3. Architecture of PAN utilized in baseline. C1, C2 and C3 represent the inputs of PAN; O1, O2 and O3 represent the outputs of PAN; UP represents nearest up-sampling with a factor of 2; and C represents feature concatenation. (Figured 3 can correctly show in word file.)

    To assess the impact of multi-scale features on detection accuracy, we visualized Grad-CAM masks for input feature maps (C3, C2, C1) and output feature maps (O3, O2, O1) of the traditional PAN in Figure 3. The detection accuracy correlates positively with the precision of information in C1, C2, and C3. Grad-CAM visualizations reveal that C1 contains more erroneous information and coherent noise compared to C2 and C3, which, if up-sampled, can propagate these inaccuracies to C2 and C3, degrading the feature fusion outcome.

    To mitigate this, we introduce SAD-FPN, a modification of the classical PAN structure that de-emphasizes high-level features during fusion, thereby reducing the transmission of coherent noise to lower-level maps. This approach significantly enhances network accuracy, as demonstrated in the ablation study in section 4.2.4.

    Additionally, Figure 4 showcases Grad-CAM masks for O1, O2, and O3, which are both the outputs of the fusion network and inputs to the head network. The focused information in O2’s Grad-CAM masks (Column (f)) significantly aids instance detection by accurately covering aircraft regions. In contrast, O3’s Grad-CAM (Column €) distributes attention evenly across the feature map, contributing less to detection. By substituting O3 with O2 as the head network input, we not only emphasize O2’s contribution but also reduce the feature map size, simplifying the network for more efficient aircraft detection in SAR scenarios. This strategy’s efficacy is confirmed in the ablation study detailed in section 4.2.4.

Figure 4. Grad-CAM masks of different scale feature maps of baseline. (a) Input image. (b) C3. (c) C2. (d) C1. € O3. (f) O2. (g) O1. (Figured 4 can correctly show in word file.)

    To elaborate on the motivations and explain how A-ELAN-H contributes to improving detection performance, the following additions have been made:

    Object detection aims to identify targets and their locations, but not all channels and features are equally important for this task. To focus on the most relevant information, we've implemented channel attention to highlight crucial channels for recognition, leveraging inter-channel relationships. This integration with the ELAN-H module strengthens the model's ability to identify targets by concentrating on significant channels. Likewise, spatial attention is used to pinpoint areas where instances are probable to occur, utilizing inter-spatial relationships to refine localization. The value of these approaches is validated by the ablation study presented in section 4.2.2.

  1. The Enhanced Hierarchical Non-Maximum Suppression (EH-NMS) algorithm seems to be a significant contribution to refining detection accuracy. Could you provide more details on how this algorithm works and its effectiveness compared to traditional NMS techniques?

A: Thank you for your valuable suggestion. We have taken it into account and made the following updates to our paper. To explain how EH-NMS works, we add following addition part:

    The EH-NMS algorithm unfolds in three primary steps. Initially, we set a threshold θ for the area ratio and initiate a loop for each image. In the second step, the bounding box with the highest score is selected as the master box (M), and the Intersection over Union (IoU) is calculated between M and the remaining candidate boxes (B). Boxes in B with an IoU exceeding the IoU threshold are eliminated, adhering to the conventional NMS approach.

    The third step is pivotal to EH-NMS. It calculates a ratio μ for each box in B by comparing the union area with M to M’s area. A μ value below θ signifies a substantial or complete overlap with the most confident detection box, leading to its exclusion. If μ exceeds θ, it indicates a minor or non-existent overlap with the most confident detection box, resulting in the retention of the box. This process of evaluating and pruning continues until all overlapping boxes are resolved.

    This refined method of NMS significantly minimizes the likelihood of overlapping detection boxes. The accompanying pseudocode provides a clear implementation guide for the EH-NMS algorithm.

Algorithm 1 EH-NMS

         Input: B = {b1, …, bn}, S = {s1, …, sn}, IoUt, θ

         B is the list of all candidate detection boxes

         S is the list of scores corresponding to all candidate boxes

         IoUt is the IoU threshold

         θ is the threshold of the ratio of the parallel area to the maximum

 possible detection frame area

         Output: K

         K indicates the detection boxes retained after the EH-NMS

1

for image in images

2

:   K ← { }

3

:   while B ≠ ø do:

4

:       m ← argmax(S)

5

:       M ← bm

6

:       K ← K ∪ M

7

:       B ← B – bm; S ← S – sm

8

:       for bi in B do:

9

:           if IoU(M, bi) < IoUt:

10

:               B ← B – bi; S ← S – si

11

:             else:

12

:               B ← B; S ← S

13

:           if ratio(union(M, bi) , area(M)) > θ:

14

:               B ← B; S ← S

15

:           else:

16

:               B ← B – bi; S ← S – si

17

:    return K

18

end

To demonstrate the effectiveness of EH-NMS, the experiment comparing with traditional NMS techniques was added as follow:

    To affirm the efficacy of the Enhanced Non-Maximum Suppression (EH-NMS) in SAR aircraft detection, we conducted a comparative analysis against four established NMS methods: Standard NMS, Soft-NMS, Distance-IoU-NMS (DIoU-NMS), and Weighted Boxes Fusion (WBF). This experiment was performed on the baseline results within the SADD dataset. Table 6 presents a comprehensive comparison matrix of these NMS techniques, highlighting the merits of EH-NMS in this context.

Table 7. Compare of multiple NMS techniques applied to the SADD baseline, utilizing an IoU threshold of 0.6.

Algorithm

AP0.5

AP0.5:0.95

Precision

Recall

NMS

89.8%

55.2%

92.6%

82.7%

Soft-NMS

85.5%

53.3%

92.6%

82.7%

DIoU-NMS

89.9%

55.3%

92.6%

82.7%

WBF

88.4%

43.5%

92.8%

80.0%

EH-NMS

90.1%

55.4%

93.0%

82.7%

    Analysis of Table 6 yields the following insights:

  1. EH-NMS excels in SAR aircraft detection by effectively eliminating highly overlapping bounding boxes, achieving superior results with 90.1% AP(0.5), 55.4% AP(0.5:0.95), and 93.0% Precision. These figures surpass the next best techniques by 0.3%, 0.1%, and 0.2%, respectively.
  2. Soft-NMS and WBF show a decline in accuracy, likely due to their optimization for natural images rather than SAR remote sensing. Meanwhile, DIoU-NMS demonstrates a slight enhancement in accuracy, indicating its robustness and generalizability.
